# Upping the Game: How 2D U-Net Skip Connections Flip 3D Segmentation

**Xingru Huang**[1][*], **Yihao Guo**[1][*], **Jian Huang**[1][*], **Tianyun Zhang**[1],
**Hong He**[1][†], **Shaowei Jiang**[1][†], **Yaoqi Sun**[1][†]
[1]Hangzhou Dianzi University

## Abstract

In the present study, we introduce an innovative structure for 3D medical image segmentation that effectively integrates 2D U-Net-derived skip connections into the architecture of 3D convolutional neural networks (3D CNNs). Conventional 3D segmentation techniques predominantly depend on isotropic 3D convolutions for the extraction of volumetric features, which frequently engenders inefficiencies due to the varying information density across the three orthogonal axes in medical imaging modalities such as computed tomography (CT) and magnetic resonance imaging (MRI). This disparity leads to a decline in axial-slice plane feature extraction efficiency, with slice plane features being comparatively underutilized relative to features in the time-axial. To address this issue, we introduce the U-shaped Connection (uC), utilizing simplified 2D U-Net in place of standard skip connections to augment the extraction of the axial-slice plane features while concurrently preserving the volumetric context afforded by 3D convolutions. Based on uC, we further present uC 3DU-Net, an enhanced 3D U-Net backbone that integrates the uC approach to facilitate optimal axial-slice plane feature utilization. Through rigorous experimental validation on five publicly accessible datasets—FLARE2021, OIMHS, FeTA2021, AbdomenCT-1K, and BTCV, the proposed method surpasses contemporary state-of-the-art models. Notably, this performance is achieved while reducing the number of parameters and computational complexity. This investigation underscores the efficacy of incorporating 2D convolutions within the framework of 3D CNNs to overcome the intrinsic limitations of volumetric segmentation, thereby potentially expanding the frontiers of medical image analysis. Our implementation is available at https://github.com/IMOP-lab/U-Shaped-Connection.

## 1 Introduction

3D volumetric data segmentation extensively relies on the utilization of axial symmetrical 3D convolutions to extract features based on a volumetric representation. Imaging modalities such as CT [1, 2, 3, 4] and MRI [5, 6, 7, 8, 9] yield high-precision images along the slice plane and repeat this process along the temporal axis, resulting in non-uniform information density across the three axes. We visualize the difference in information density between the axial and the slice plane in 3D medical imaging in Fig. 1. The information density variance arising from non-simultaneous imaging in three dimensions of volumetric data engenders distinctiveness between time-axial and slice plane features, rendering them unsuitable for the same processing. Typically, features extracted from the slice plane possess higher information density and are more adept at delineating the fine-grained, voxel-level neighboring structures. These localized features are essential in understanding the precise 3D structure of tissues, thereby surpassing the utility of time-axial features. However,

---

[*]These authors contributed equally.

[†]Corresponding author. Email: hehong@hdu.edu.cn, jiangsw@hdu.edu.cn, syq@hdu.edu.cn

38th Conference on Neural Information Processing Systems (NeurIPS 2024).

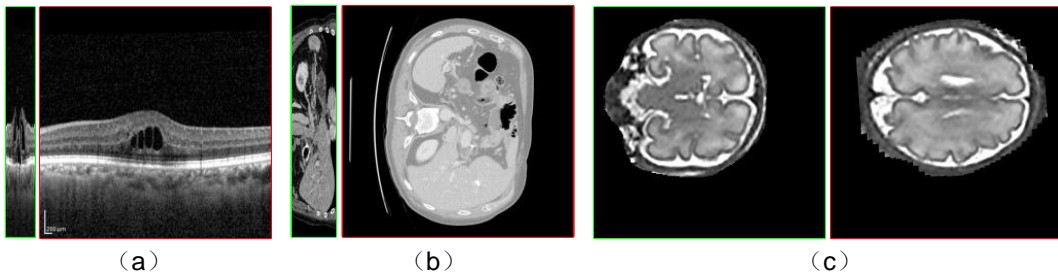

（a）        （b）        （c）

Figure 1: Visualization of the difference in information density between the time-axial (Green) and slice planes (Red). Panels (a), (b), and (c) respectively illustrate the information density differences in time-axial and slice planes for volumetric data of the abdomen, retina, and brain tissues.

traditional 3D convolutions [10, 11, 12], although excelling in time-axial information extraction, significantly increase computational complexity to accommodate time-axial information, leading to axial-slice plane performance drop-off in 3D CNNs. Thus, the structural constraints of 3D convolutions inherently weaken the rich local features of the slice plane. When confronted with sequences abundant in slice plane features, symmetric 3D convolutions face substantial inefficiencies and performance degradation.

To address the axial-slice plane performance drop-off issue, an efficient strategy involves increasing channel depth to enhance feature extraction performance. The depth of the channels, indicative of the high-dimensional feature richness, directly influences network performance. Increasing channel depth can enhance the network's capacity to capture more high-dimensional information and improve feature representation quality. However, the relationship between channel depth and network performance in 3D CNNs is intricate and nonlinear. While initially increasing channel depth improves model performance by introducing more detailed and abstract features, this improvement comes at the expense of ever-increasing computational overhead. Moreover, an excessive number of channel depths can complicate the model's information processing, potentially introducing noise and redundancy, culminating in dimensional explosion and performance decline. The linear relationship between channel depths and computational overhead means that increasing channel depth significantly inflates computational costs, leading to reduced model efficiency under identical hardware constraints. Conversely, 2D convolutions, with their focus on slice-wise information extraction, exhibit higher efficiency and performance peaks, boasting a superior parameter performance ratio. For a kernel size $K$, a 2D convolution requires only $\frac{1}{K}$ of the parameters compared to its 3D counterpart, thereby significantly enhancing local feature extraction efficiency within a slice. Consequently, 2D convolutions present a key solution to the low parameter-to-performance ratio of 3D CNN architectures, markedly increasing axial-slice plane feature extraction efficiency while minimizing parameter count and computational load.

Given the excellent computational efficiency and parameter-to-performance ratio of 2D convolutions for axial-slice plane feature utilization, and considering the prevalent use of skip connections in 3D medical image segmentation architectures to integrate early layer details, we propose employing 2D convolutions within skip connections to supplement axial-slice plane information. For this purpose, we introduce a plug-and-play U-shaped Connection (uC), leveraging a simplified 2D U-Net to replace skip connections in 3D segmentation architectures for improving the utilization efficiency of axial-slice plane features. Using classical 3D U-Net as the backbone, we further propose the uC 3DU-Net, which substitutes the original skip connections with uC, thereby enhancing the network's ability to comprehend slice plane information and efficiently utilize initial encoded layer details. For more effective integration of 3D spatial features from 3D CNNs and 2D slice plane features introduced by uC, we employ a Dual Feature Integration (DFi) module to combine multi-dimensional features.

Experimental evaluations demonstrate the incorporation of uC significantly enhances segmentation performance even with reduced channels and surpasses parameter-to-performance ratios of state-of-the-art networks. Comparative analyses on five publicly available datasets—FLARE2021, OIMHS, FeTA2021, AbdomenCT-1K, and BTCV—corroborate the superiority of proposed uC 3DU-Net over existing state-of-the-art methods, achieving persuasive performance with reduced computational complexity and model parameters.

The principal contributions of our work can be summarized as follows:

1. We proposed uC, utilizing simplified 2D U-Net to replace traditional skip connections in 3D segmentation backbones, enhancing model capacity to capture axial-slice plane features.

2. uC 3DU-Net is further proposed, adopting uC to replace original skip connections, and applying the DFi module to effectively merge 3D sequential spatial features with 2D axial-slice plane features, thus improving feature utilization efficiency.

3. We explore the complementary relationship between uC and 3D CNNs, revealing that incorporating 2d convolutions in 3D CNNs can achieve superior performance with fewer parameters, striking perfect balances between efficiency and performance in volumetric segmentation.

4. Comparative experiments on five public datasets—FLARE2021, OIMHS, FeTA2021, AbdomenCT-1K, and BTCV—demonstrate that uC 3DU-Net surpasses all previous models, achieving SOTA performance with fewer parameters and lower computational cost.

## 2 Related Work

### 2.1 3D medical image segmentation

The realm of 3D medical image segmentation has experienced substantial advancements, primarily propelled by the progressive evolution of deep learning paradigms and the augmentation of computational prowess [13, 14]. At the key of these advancements lies the continuous enhancement of CNNs [15, 16, 17, 18, 19] and the advent of Vision Transformers (ViTs) [20], both of which have significantly promoted segmentation methodologies [21, 22, 23, 24, 25]. The 3D U-Net [26], a foundational model in this field, the subsequent iterations have incorporated attention mechanisms, augmenting feature extraction capabilities by prioritizing important regions while mitigating noise interference [27, 28]. Transformer-based architectures, exemplified by the UNETR framework [29, 30], leverage multi-head self-attention mechanisms to capture long-range dependencies, thus facilitating more precise and meticulous segmentation. Hybrid models that combine CNNs with Transformers, such as TransUNet [31] and TransBTS [32], effectively balance local spatial feature extraction with global contextual understanding, culminating in superior segmentation performance [33, 34]. The nnU-Net [35], an exemplary self-adaptive framework, has demonstrated exceptional performance across a diverse array of medical imaging tasks by autonomously configuring itself to specific datasets.

Beyond architectural innovations, a multitude of research endeavors have concentrated on refining 3D medical image segmentation mask techniques. Chen et al. [36] propose a novel method integrating Active Appearance Models (AAM) with live wire (LW) and Graph Cuts (GC) techniques, thereby enhancing the efficacy of 3D medical image segmentation. Zhang et al. [21] develop the 3D Context Residual Network (ConResNet), which employs a context residual module to interlink the segmentation decoder with the context residual decoder, explicitly learning inter-slice contextual information to improve segmentation accuracy. Advanced loss functions and optimization strategies, such as extensive implementations of Dice loss [37], have been designed to address class imbalance issues, ensuring a more balanced learning process between easy and hard examples. Furthermore, data augmentation and transfer learning techniques have been beneficial in enhancing model robustness and generalization capabilities [38, 39]. These unremitting innovations promise increasingly precise and reliable analysis of 3D medical images, significantly advancing the fields of medical diagnostics and therapeutic planning.

### 2.2 Advancements in skip connection structures

Innovations in skip connection structures have further augmented the performance and computational efficiency of CNNs [40, 41, 42] in the domain of medical image segmentation. Skip connections are important in facilitating direct gradient propagation, thereby mitigating the vanishing gradient issue and enabling the training of substantially deeper networks. This method permits unimpeded gradient flow throughout the network, effectively circumventing intermediate layers, minimizing information degradation, and achieving more stable and efficient optimization. Prominently exemplified in the ResNet[43] architecture, skip connections mitigate the vanishing gradient dilemma, thereby fostering the training of deep neural networks. In the U-Net[44] architecture, skip connections are crucial in the transmission of initial detailed spatial information from the encoder to the decoder, thereby preserving high-resolution details indispensable for precise segmentation tasks.

Recent endeavors have extensively refined traditional skip connections to augment their functional efficacy. Attention U-Net [45] replaces the normal application of skip connections by incorporating attention mechanisms that dynamically emphasize important features while attenuating irrelevant information. Dense U-Net [46] leverages densely connected convolutional networks (DenseNet) [47] to facilitate feature reuse and improve gradient flow. Interconnecting each layer with every other layer in a feed-forward manner facilitates the assimilation of richer and more diverse feature representations. Skip connections in the Residual Channel Attention Network (RCAN) [48] network mitigate the training difficulty of deep neural networks and enhance feature reuse and information flow efficiency, thereby improving the network's ability to learn high-frequency information. Hybrid Densely Connected UNet (H-DenseUNet) [49] incorporates skip connections within dense blocks, combining the strengths of DenseNet and U-Net. By harnessing advanced feature reuse, optimized gradient flow facilitated by dense connectivity, and efficient multi-scale feature fusion, it produces a robust architecture adept at capturing intricate details and nuanced contextual information.

## 3   Method

### 3.1   Preliminaries: 3D convolutional segmentation networks and skip connections

Current 3D medical image segmentation networks primarily utilize architectures based on 3D convolutions, as exemplified by classic models such as 3D U-Net, SegResNet, and the recent 3D UX-Net. These networks adopt an end-to-end encoder-decoder framework with skip connections. The encoder progressively increases channel depth while downsampling spatial feature maps to abstract and consolidate contextual information, thereby reducing parameter complexity and improving computational efficiency. In contrast, the decoder gradually upsamples these encoded features to restore the original resolution, refining segmentation boundaries and overall segmentation accuracy. Skip connections effectively reintroduce fine-grained details from the original images during the upsampling stage, aiding the refinement of segmentation results.

Several 3D medical image segmentation networks, such as UNETR, TransBTS, and SwinUNETR, have incorporated multi-head self-attention and advanced skip connections within the 3D Conv-based encoder-decoder framework. These hybrid approaches effectively capture long-range dependencies and facilitate multi-scale feature utilization. Multi-head self-attention ensures global consistency, while 3D convolution operations excel in preserving local spatial details. The combination of self-attention with 3D convolutions ensures the model's ability to retain image details while comprehending and processing global information more effectively. However, relying solely on transformer architectures, such as Vision Transformers, would result in a substantial increase in parameters and computational load. Additionally, without skip connections, it is difficult to achieve satisfactory segmentation details. Hence, 3D convolutions and skip connections remain crucial for achieving optimal segmentation performance in 3D medical image segmentation.

### 3.2   U-shaped Connection

Considering the objective of skip connections to capture detailed features of original images, it is essential to note that 3D medical images fundamentally represent a sequence of 2D images. Given the anisotropic nature of medical images, the 2D slice plane shows richer feature information compared to the temporal sequence axis. Although basic skip connection methods like cat and addition can supplement original feature information to some extent, these methods fail to fully utilize the rich axial-slice plane information in 3D medical images. To address the challenges posed by the anisotropy of 3D medical images, the U-shaped Connection(uC) is proposed. It offers a solution to anisotropy in medical imaging. This approach employs a simplified 2D U-Net to implement skip connections, thereby supplementing the rich original 2D slice plane feature information, and enhancing the 2D slice plane feature extraction capabilities of any 3D image segmentation network. The uC combines features extracted by 2D convolution with those extracted by 3D upsampling, offering more efficient feature extraction compared to pure 3D convolution, as detailed structure in Sec3.3.

The fundamental structure of uC is based on a 2D U-Net. In comparison to the basic 2D U-Net, uC omits the initial and final conv layers, retaining only the downsampling and upsampling layers to minimize computational load and enhance the extraction efficiency of original slice plane features. Each downsampling layer comprises an average pooling layer and two conv layers, each consisting of 2D conv, Group Normalization (GN), and ReLU. The average pooling layer reduces the feature map size by half, and the subsequent conv layers double the channel number, with the group number of GN set to half the current channel number. Each upsampling layer includes a transposed convolution layer and two conv layers, where the transposed convolution layer restores the feature map to its original size, and the conv layers reduce the channel number back to its half.

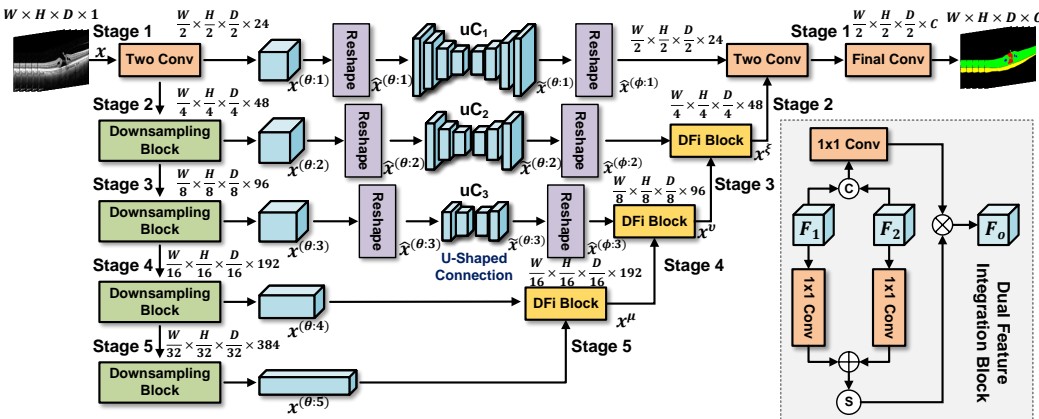

Figure 2: **Overview of the proposed uC 3DU-Net architecture.** The backbone is a 3D U-Net with five encoding-decoding stages and each downsampling block comprises a max-pooling layer followed by two 3D convolutional layers. In stages 1 to 3, 5D tensors are rearranged into 4D tensors for 2D uC input by stacking slices along the batch dimension. Each upsampling layer employs a transposed convolution to upsampling and a DFi block to effectively adjust the feature channel depth.

### 3.3 uC 3DU-Net

Based on uC, we proposed uC 3DU-Net, which employs the 3D U-Net as the backbone, incorporating uC with a dual feature integration (DFi) module. The network architecture is illustrated in Fig. 2. Given the input 3D volumetric image data denoted as $x$, we divide the encoder of the 3D U-Net into five stages. Stage 1 is the initial Two Conv layer, comprising two 3D convolutions, expanding the input feature channel depth to 24. Stages 2 to 5 are downsampling layers, each comprising a max-pooling operation and two 3D convolutions, doubling the feature channels while halving the feature map dimensions. The outputs of stages 1 to 5 are denoted as $x^{(\theta:1)}$ to $x^{(\theta:5)}$. The decoder is also divided into five stages. Stages 5 to 2 utilize transposed convolutions to upsample, effectively doubling the spatial dimensions of the feature maps. Stages 5 to 3 apply DFi to effectively halve the feature channel depth, while Stage 2 retains two convolution layers. Stage 1 is the Final Conv layer, converting the feature channels to the number of categories by a 3D convolution. All 3D convolution operations are subsequently followed by Instance Normalization and a LeakyReLU activation function, with the negative slope set to 0.1.

Considering the parameter-efficiency trade-off, we employ the uC in only stages 1 to 3, denoted as $uC_1$ to $uC_3$. The inputs $x^{(\theta:1)}$ to $x^{(\theta:3)}$ are reshaped into 4D dimensions by stacking slices along the batch dimension using a rearrange operation, resulting in $\hat{x}^{(\theta:1)}$ to $\hat{x}^{(\theta:3)}$. The structure of $uC_1$ to $uC_3$ resembles that of a simplified version of 2D U-Net, with detailed descriptions of the uC architecture provided in Section 3.2. $uC_1$, $uC_2$, and $uC_3$ consist of 4, 3, and 2 downsampling and upsampling layers, respectively, to match the dimensions of the input 4D tensor. The outputs of $\hat{x}^{(\theta:1)}$ to $\hat{x}^{(\theta:3)}$ after passing through the uC are $\tilde{x}^{(\theta:1)}$ to $\tilde{x}^{(\theta:3)}$, which are reshaped back to 5D dimensions using a rearrange operation, resulting in $\hat{x}^{(\phi:1)}$ to $\hat{x}^{(\phi:3)}$.

We utilize the dual feature integration (DFi) module to seamlessly integrate the 3D spatial features extracted by the 3D CNN with the rich axial-slice plane features introduced by the uC. Specifically, $x^{(\theta:4)}$ and $x^{(\theta:5)}$ are fused using the DFi block to obtain $x^{\mu}$, $x^{\mu}$ and $\hat{x}^{(\phi:3)}$ are fused to obtain $x^{\upsilon}$, and $x^{\upsilon}$ and $\hat{x}^{(\phi:2)}$ are fused to obtain $x^{\xi}$. Finally, $x^{\xi}$ and $\hat{x}^{(\phi:1)}$ are fused using the DFi block and undergo the final convolutional block to obtain the final output $y$.

**Dual Feature Integration.** In conventional 3D U-Net architectures, skip connections integrate the original feature information from the encoder during the upsampling process through cat, followed by two 3x3 convolutions to reduce the number of channels and consolidate useful information. However, this basic feature fusion approach may lead to inefficiencies when reconciling the disparity between 3D spatial features and 2D slice plane features. Therefore, we devise a streamlined, efficient multi-scale feature fusion module DFi to better integrate the axial-slice plane information introduced by uC. The 2D slice plane features extracted by uC and the 3D spatial features from the 3D CNN

encoder are cat to retain maximal original feature information. However, subsequent convolution operations to reduce channel depths are computationally intensive and inefficient. Utilizing addition for feature fusion could result in information loss due to inherent discrepancies between the two types of features. Hence, an approach combining cat, addition, and subsequent convolution layers, with the aim of minimizing parameter count and FLOPs, resulting in the design of the DFi module. For the 2D slice plane features F1 extracted by uC and the 3D spatial features F2 obtained during 3D U-Net upsampling, branch 1 of the DFi module cats F1 and F2, then employs a 1x1 conv to halve the channel count, preserving important features to produce feature map Fc. Branch 2 initially extracts critical information from F1 and F2 using 1x1 convs, followed by addition and sigmoid activation to generate the attention map Fa. The element-wise multiplication of Fc and Fa adjusts and integrates the feature map Fc from branch 1, thereby learning the relative importance of F1 and F2 during the fusion process. The proposed DFi module integrates and simplifies addition, cat, and subsequent convolution layers, achieving a more lightweight and efficient fusion of 2D and 3D spatial features.

# 4 Experiments and results

## 4.1 Datasets

To validate the efficacy of the proposed method, multiple experiments are conducted on five widely-used, publicly accessible datasets: FLARE2021 [50], OIMHS [51], FeTA2021 [52], AbdomenCT-1K [53], and BTCV [54]. To ensure experimental rigor and fairness, we apply identical data preprocessing and hyperparameter settings across all datasets, uniformly utilizing the $\mathcal{L}_{DiceCE}$ for training.

**FLARE2021.** An anisotropic CT dataset dedicated to abdominal organ segmentation comprises 361 training instances, 50 validation instances, and 100 test instances, categorized into four organ classes: spleen, kidney, liver, and pancreas. The spatial resolution ranges from 0.61 mm to 0.98 mm in-plane and 0.5 mm to 7.5 mm through-plane. We utilize the full set of 361 publicly labeled instances.

**OIMHS.** A fundus retinal 3D OCT segmentation dataset comprises 125 sequences, each containing 19 to 73 scans, and is categorized into four classes: retina, choroid, macular hole, and macular edema. It is a anisotropic dataset, the spacing ranges from 10.7 μm to 14.0 μm in-plane and 7.0 μm to 40.0 μm through-plane. All 125 publicly available sequences are utilized in the experiments.

**FeTA2021.** A dataset consists of 120 T2-weighted fetal brain MRI reconstructions, categorized into seven classes: external cerebrospinal fluid, grey matter, white matter, ventriculus, cerebellum, deep grey matter, and brainstem. It is a typical isotropic dataset, the spacing ranges from 0.43 mm to 1 mm across all dimensions. All 80 publicly available sequences are utilized in the experiment.

**AbdomenCT-1K.** A large-scale abdominal CT dataset comprises 1,112 instances, focusing on the segmentation of four abdominal organs: liver, kidney, spleen, and pancreas. The spatial resolution ranges from 0.45 mm to 1.04 mm in-plane and 0.45 mm to 8 mm through-plane. It is also an anisotropic dataset, for our experiments, we utilize 361 instances from Task 2.

**BTCV.** A dataset consists of 50 abdominal CT instances, categorized into 13 classes. It is an anisotropic dataset, with in-plane resolution ranging from 0.59 mm to 0.98 mm and through-plane resolution ranging from 2.5 mm to 5.0 mm. All 30 labeled instances are selected for the experiments.

## 4.2 Implementation details and evaluation metrics

**Implementation details.** The datasets are randomly partitioned in an 8:1:1 ratio for training, validation, and testing. To enhance segmentation performance, the data are resampled, intensity clipped, and normalized using min-max normalization. Specifically, the intensity clipping ranges are as follows: FLARE2021 $[-125, 275]$, OIMHS $[0, 300]$, FeTA $[0, 1000]$, AbdomenCT-1K $[-125, 300]$, and BTCV $[-175, 250]$. During the training stage, random cropping to $96 \times 96 \times 96$ volumes is performed, and a sliding window method with a 0.5 overlap is used for validation.

All training utilizes the $\mathcal{L}_{DiceCE}$ function with the AdamW [55] optimizer, a learning rate of 0.0001, 80,000 training iterations, and a batch size of 2. Data augmentation techniques, including random flip, random rotation, random scaling, and random 3D elastic transformation, are applied to enhance dataset diversity and model generalization. Validation occurs every 1000 iterations during training to save the model weight with the best validation performance. The experiments are conducted on identical hardware and software environments, each workstation equipped with two NVIDIA

Table 1: Comparative experimental results of uC 3DU-Net and 9 previous methods on the FLARE2021 and FeTA2021 datasets. The best values for each metric are bolded. Part of the data comes from [56].

| Methods | #Params | FLOPs | FLARE2021 | | | | | FeTA2021 | | | | | | | |
|---|---|---|---|---|---|---|---|---|---|---|---|---|---|---|---|
| | | | Mean | Spleen | Kidney | Liver | Pancreas | Mean | ECF | GM | WM | Vent | Cereb | DGM | BS |
| 3D U-Net [57] | 4.81M | 135.9G | 89.2 | 91.1 | 96.2 | 90.5 | 78.9 | 85.7 | 86.7 | 76.2 | 92.5 | 86.1 | 91.0 | 84.5 | 82.7 |
| SegResNet [58] | 1.18M | 15.6G | 90.2 | 96.3 | 93.4 | 96.5 | 74.5 | 86.2 | 86.8 | 77.0 | 92.7 | 86.5 | 91.1 | 86.7 | 82.5 |
| RAP-Net [59] | 38.2M | 101.2G | 91.3 | 94.6 | 96.7 | 94.0 | 79.9 | 86.5 | 88.0 | 77.1 | 92.7 | 86.2 | 90.7 | 87.9 | 83.2 |
| nn-UNet [35] | 31.2M | 743.3G | 92.6 | 97.1 | 96.6 | 97.6 | 79.2 | 87.0 | 88.3 | 77.5 | 93.0 | 86.8 | **92.0** | 88.0 | 84.0 |
| TransBTS [32] | 31.6M | 110.4G | 90.2 | 96.4 | 95.9 | 97.4 | 71.1 | 86.8 | **88.5** | 77.8 | 93.2 | 86.1 | 91.3 | 87.6 | 83.7 |
| UNETR [29] | 92.8M | 82.6G | 88.6 | 92.7 | 94.7 | 96.0 | 71.0 | 86.0 | 86.1 | 76.2 | 92.7 | 86.2 | 90.8 | 86.8 | 83.4 |
| nnFormer [60] | 149.3M | 240.2G | 90.6 | 97.3 | 96.0 | 97.5 | 71.7 | 86.3 | 88.0 | 77.0 | 93.0 | 85.7 | 90.3 | 87.6 | 82.8 |
| SwinUNETR [34] | 62.2M | 328.4G | 92.9 | 97.9 | 96.5 | 98.0 | 78.8 | 86.7 | 87.3 | 77.2 | 92.9 | 86.9 | 91.4 | 87.5 | 84.2 |
| 3D UX-Net [56] | 53.0M | 639.4G | 93.4 | **98.1** | **96.9** | 98.2 | 80.1 | 87.4 | 88.2 | 78.0 | **93.4** | 87.2 | 91.7 | **88.6** | 84.5 |
| **uC 3DU-Net** | 21.7M | 286.4G | **94.1±1.61** | 98.0±1.02 | 96.7±1.45 | **98.3±0.76** | **83.2±5.37** | **87.8±1.99** | 87.8±2.57 | **80.7±2.29** | 92.6±1.89 | **89.1±3.52** | 91.0±2.47 | 86.3±5.01 | **87.5±2.32** |

Table 2: Comparative experimental results of uC 3DU-Net and 7 previous methods on the OIMHS dataset. The best values for each metric are highlighted in bold. Part of the data comes from [25].

| Method | #Params | FLOPs | mIoU | Dice | VOE | HD95 | AdjRand |
|---|---|---|---|---|---|---|---|
| 3D U-Net [57] | 4.81M | 135.9G | 86.02 | 92.05 | 13.98 | 6.77 | 91.34 |
| Swin UNETR [34] | 62.2M | 328.4G | 86.73 | 92.53 | 13.27 | 5.09 | 91.85 |
| 3D UX-Net [56] | 53.0M | 639.4G | 87.43 | 92.90 | 12.57 | 4.41 | 92.27 |
| SASAN [61] | 22.96M | 282.92G | 88.44 | 93.53 | 11.56 | 3.14 | 92.96 |
| nnFormer [60] | 149.3M | 240.2G | 72.16±7.91 | 81.60±7.41 | 27.84±25.07 | 23.49±7.91 | 80.36±7.73 |
| TransBTS [32] | 31.6M | 110.4G | 74.80±7.31 | 83.08±6.55 | 25.20±23.82 | 31.43±7.31 | 82.05±6.85 |
| UNETR [29] | 92.8M | 82.6G | 80.52±6.68 | 88.11±5.41 | 19.48±30.31 | 30.07±6.68 | 87.21±5.56 |
| **uC 3DU-Net** | 21.7M | 286.43G | **89.48±3.56** | **94.13±2.56** | **10.52±5.61** | **2.98±3.56** | **93.62±2.66** |

Table 3: Comparative experimental results of uC 3DU-Net and 6 previous methods on the AbdomenCT-1K dataset. The best values for each metric are highlighted in bold.

| Method | #Params | FLOPs | mIoU | Dice | ASSD | HD95 | AdjRand |
|---|---|---|---|---|---|---|---|
| 3D U-Net [57] | 4.81M | 135.9G | 86.69±4.30 | 92.28±2.93 | 2.31±2.06 | 12.68±18.54 | 92.14±2.98 |
| TransBTS [32] | 31.6M | 110.4G | 70.06±9.11 | 79.64±8.12 | 4.88±1.59 | 35.75±10.00 | 79.35±8.19 |
| UNETR [29] | 92.8M | 82.6G | 84.17±5.12 | 90.42±3.75 | 2.36±1.68 | 13.41±16.77 | 90.27±3.81 |
| nnFormer [60] | 149.3M | 240.2G | 80.69±7.76 | 87.56±6.36 | 2.01±1.80 | 9.75±9.95 | 87.38±6.41 |
| Swin UNETR [34] | 62.2M | 328.4G | 86.76±4.94 | 92.36±3.20 | 2.67±1.81 | 14.87±18.78 | 92.22±3.26 |
| 3D UX-Net [56] | 53.0M | 639.4G | 86.56±4.76 | 92.21±3.18 | 2.60±1.98 | 15.46±19.75 | 92.07±3.24 |
| **uC 3DU-Net** | 21.7M | 286.4G | **88.29±4.04** | **93.35±2.60** | **1.48±1.09** | **8.53±10.74** | **93.22±2.66** |

GeForce RTX 4090 GPUs and 128GB of memory. The framework employs Python 3.9, PyTorch 2.0.0, and MONAI 0.9.0 within a Distributed Data-Parallel (DDP) training framework.

**Evaluation metrics.** We utilize IoU/mIoU, Dice, ASSD, HD, HD95, VOE, and AdjRand as evaluation metrics to comprehensively assess segmentation performance. IoU and mIoU measure the overlap accuracy between predicted and ground truth regions, providing robustness across multiple classes. Dice is particularly effective for medical images, especially in small target regions. ASSD calculates average surface distances, while HD and HD95 assess boundary accuracy, with HD95 focusing on the 95th percentile to reduce outliers. VOE examines volumetric overlap and AdjRand evaluates clustering similarity, ensuring a thorough assessment of both overlap and boundary precision.

### 4.3 Comparison with state-of-the-art methods

Comparative experiments are conducted on proposed uC 3DU-Net and previous state-of-the-art methods across five diverse, publicly available 3D medical image segmentation datasets: FLARE2021, OIMHS, FeTA2021, AbdomenCT-1K, and BTCV. The results are presented in Table 1, Table 2, Table 3, and Table 13 (provided as supplementary material in the Appendix). The FLARE2021, FeTA2021, and OIMHS datasets are evaluated using five-fold cross-validations. The results from all five folds pass the Wilcoxon signed-rank test with a significance level of $\alpha=0.05$, indicating statistical equivalence in data distributions. Additionally, the results from the AbdomenCT-1K dataset all pass the one-tailed Wilcoxon signed-rank test with $p<0.05$, thereby validating the SOTA performance of the uC 3D U-Net. Overall, the result indicates that the proposed uC 3DU-Net achieved SOTA performance with fewer model parameters and reduced computational complexity compared to previous segmentation methods. On the FLARE2021 and FeTA datasets, uC 3DU-Net's parameter count and FLOPs are only 40.9% and 44.7% of the suboptimal model, 3D UX-Net, yet the average Dice scores improve by 0.7% and 0.4%, respectively, with a notable 3.1% increase in the pancreas category. On the OIMHS and AbdomenCT-1K datasets, uC 3DU-Net outperformed previous models by at least 0.6% and 0.99% in Dice scores, respectively, highlighting its superior overall segmentation performance. Additionally, uC 3DU-Net showed significant improvements in boundary performance metrics such as ASSD and HD95 on the OIMHS and AbdomenCT-1K datasets, demonstrating its enhanced ability to identify and correct boundary pixels by focusing on axial-slice plane features. These findings substantiate that the 2D Conv-based uC effectively compensates for the limitations of 3D convolutions in extracting in-plane slice information to enhance model performance. Notably, the performance improvement observed on the anisotropic datasets surpasses that on the isotropic FeTA dataset, further highlighting the efficiency of the proposed uC in enhancing in-slice plane information extraction, especially for anisotropic datasets enriched with axial-slice plane features. Overall, uC significantly improves in-plane feature extraction efficiency while maintaining robust axial spatial feature representation, representing the pinnacle of current performance and parameter efficiency.

### 4.4 Analysis

#### 4.4.1 Evaluation of 2D vs. 3D convolution for slice-plane information extraction

**Image Reconstruction Fidelity Comparison.** We evaluate three methods: 3D U-Net, 3D U-Net + 3D uC, and 3D U-Net + 2D uC, on their ability to reconstruct axial-slice plane images from input 3D sequence images, with the results presented in Table 4. The highest PSNR achieved by 3D U-Net + 2D uC confirms the proposed 2D uC's superior slice-plane feature extraction and reconstruction abilities compared to 3D Conv.

Table 4: PSNR results of the reconstruction experiments on the OIMHS dataset, including three methods: 3D U-Net, 3D U-Net + 3D uC, and 3D U-Net + 2D uC.

|  | 3D U-Net(32) | +3D uC | +2D uC |
|---|---|---|---|
| PSNR | 28.98db | 34.93db | **36.29db** |

**Impact of Convolution Dimensionality on uC Performance and Parameters.** We conduct experiments on the OIMHS dataset using 3D U-Net, 3D U-Net + 3D uC, and 3D U-Net + 2D uC across varying channel depths to evaluate the impact of 2D and 3D convolutions on uC performance and parameter count. As shown in Table

Table 5: Results of 3D U-Net, 3D U-Net + 3D uC, and 3D U-Net + 2D uC across different channel depth numbers on the OIMHS dataset. The best values for each metric are highlighted in bold.

| Methods | #Params | FLOPs | mIoU | Dice | ASSD | HD95 | AdjRand |
|---|---|---|---|---|---|---|---|
| 3D U-Net (8) | 0.37M | 16.50G | 83.17±4.51 | 90.20±3.20 | 2.88±2.16 | 16.92±21.89 | 89.51±3.23 |
| +3D uC | 5.56M | 46.80G | 85.58±3.60 | 91.84±2.35 | 2.25±1.35 | 8.64±14.07 | 91.19±2.40 |
| +2D uC | 2.85M | 42.54G | **89.59±2.61** | **94.31±1.62** | **0.52±0.48** | **2.57±4.74** | **93.84±1.65** |
| 3D U-Net (16) | 1.47M | 65.09G | 87.08±3.43 | 92.77±2.24 | 0.92±0.92 | 4.57±8.92 | 92.20±2.29 |
| +3D uC | 22.21M | 186.27G | 87.50±3.34 | 93.08±2.11 | 4.61±1.25 | 43.45±19.56 | 92.41±2.15 |
| +2D uC | 11.38M | 169.20G | **89.98±3.43** | **94.50±2.23** | **0.39±0.42** | **2.56±4.96** | **94.06±2.26** |
| 3D U-Net (32) | 4.81M | 135.90G | 87.08±3.32 | 92.79±2.09 | 1.31±1.35 | 7.39±14.69 | 92.22±2.15 |
| +3D uC | 88.67M | 596.06G | 87.93±3.33 | 93.32±2.09 | 3.09±1.31 | 14.13±16.99 | 92.73±2.12 |
| +2D uC | 45.33M | 527.75G | **90.52±2.77** | **94.84±1.68** | **0.35±0.33** | **2.20±4.07** | **94.43±1.76** |

5 and Fig. 5a (provided as supplementary material in the Appendix along with Fig. 5b), the 2D uC consistently achieves a superior parameter-to-performance ratio. This performance difference, independent of skip connections, is directly attributed to the differing feature extraction efficiencies of the 2D and 3D convolutions within the uC structure. This result is consistent with Table 4, further confirming the effectiveness of 2D convolutions for axial-slice plane feature extraction. Moreover, Fig. 5b indicates that the 2D uC converges faster and achieves a higher performance ceiling in the validation set, highlighting its greater performance in axial-slice plane feature extraction.

### 4.4.2 Impact of uC on various 3D segmentation backbones

To assess uC's efficacy in overcoming axial-slice plane performance drop-off, we conducted quantitative analysis experiments on the FLARE2021 dataset using numerous state-of-the-art models. As depicted in Table 6 and Fig. 3, replacing traditional skip connections with uC can significantly

Table 6: Analysis experiments to evaluate the effectiveness and robustness of the uC across different backbones on the FLARE2021.

| Method | #Params | FLOPs | mIoU | Dice | ASSD | HD95 | AdjRand |
|---|---|---|---|---|---|---|---|
| SegResNet | 18.8M | 244.4G | 88.38±2.07 | 93.39±1.35 | 1.31±1.29 | 6.37±10.45 | 93.28±1.37 |
| **SegResNet +uC** | 20.5M | 426.7G | **89.42±1.71** | **94.04±1.05** | **0.80±0.45** | **2.35±1.03** | **93.94±1.08** |
| TransBTS | 31.6M | 110.4G | 87.63±2.74 | 92.84±1.86 | 1.04±0.64 | 3.54±2.64 | 92.73±1.88 |
| **TransBTS +uC** | 36.6M | 176.4G | **89.46±2.03** | **94.05±1.32** | **0.78±0.30** | **2.51±1.17** | **93.95±1.34** |
| SwinUNETR | 62.2M | 328.4G | 88.28±2.69 | 93.23±1.83 | 0.95±0.41 | 3.25±2.18 | 93.13±1.85 |
| **Swin UNETR +uC** | 75.4M | 612.2G | **89.07±2.60** | **93.72±1.83** | **0.76±0.41** | **2.56±1.14** | **93.62±1.85** |
| 3D UX-Net | 53.0M | 639.4G | 88.62±2.90 | 93.43±1.94 | 2.59±6.00 | 13.27±46.73 | 93.32±2.00 |
| **3D UX-Net +uC** | 60.6M | 867.91G | **89.62±2.31** | **94.11±1.52** | **0.76±0.35** | **2.53±1.34** | **94.01±1.55** |

enhance axial-slice feature extraction capabilities and thereby improve overall performance. Incorporating uC exhibited improvements in mIoU and Dice scores by 0.79%-1.83% and 0.65%-1.21%, respectively. Integrating uC into high-performing 3D medical image segmentation models such as 3D UX-Net, which initially achieved a 93.43% Dice score, resulted in a 1.0% mIoU and 0.77% Dice score improvement. This demonstrates that even state-of-the-art models encounter inherent axial-slice plane performance drop-off issues, which uC effectively mitigates, thereby enhancing axial-slice plane feature extraction and overall performance.

### 4.4.3 Analysis on DFi

To validate the efficiency of the DFi module compared to the two 3x3 convolutions in uC 3DU-Net for integrating 2D and 3D features, we conduct analysis experiments on the OIMHS dataset to evaluate the impact of incorporating DFi on the performance of uC 3DU-Net(32), considering different quantities and positions of uC

Table 7: Further analysis experiments on DFI are conducted to validate its efficiency and effectiveness on the OIMHS dataset. The best values for each metric are highlighted in bold.

| Methods | #Params | FLOPs | mIOU | Dice | ASSD | HD95 | AdjRand |
|---|---|---|---|---|---|---|---|
| uC(stage1) | 30.38M | 439.87G | 90.06±3.41 | 94.55±2.16 | 0.96±1.85 | 7.93±19.51 | 94.14±2.20 |
| +DFi | 23.58M | 377.19G | **90.42±2.59** | **94.79±1.56** | **0.68±1.19** | **4.45±11.03** | **94.38±1.62** |
| uC(stage2) | 30.28M | 330.01G | 89.73±3.12 | 94.37±1.92 | **0.35±0.38** | **2.36±4.69** | 93.94±1.99 |
| +DFi | 23.49M | 267.34G | **90.03±3.09** | **94.53±1.93** | 0.49±0.50 | 2.50±5.19 | **94.11±1.99** |
| uC(stage3) | 29.92M | 286.09G | 88.71±3.31 | 93.75±2.12 | **0.43±0.42** | 2.70±5.17 | 93.26±2.18 |
| +DFi | 23.12M | 223.39G | **89.33±3.30** | **94.13±2.07** | 0.55±0.64 | **2.67±5.30** | **93.68±2.13** |
| uC(stage1,2,3) | 45.33M | 527.75G | 90.52±2.77 | 94.84±1.68 | 0.35±0.33 | **2.20±4.07** | 94.43±1.76 |
| +DFi | 38.53M | 465.07G | **90.86±2.75** | **95.03±1.68** | **0.34±0.45** | 2.34±5.23 | **94.63±1.73** |

module. As presented in Table 7, the results indicate that DFi achieves a slight performance improvement with lower parameter and FLOPs counts than direct concatenation. This improvement

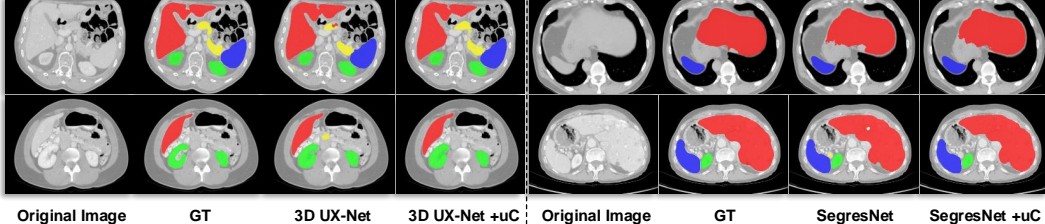

| Original Image | GT | 3D UX-Net | 3D UX-Net +uC | Original Image | GT | SegresNet | SegresNet +uC |

Figure 3: Qualitative results of the uC's impact on segmentation performance in 3D UX-Net and SegResNet backbones, applied to the FLARE2021 dataset. Segmentation results for different categories are represented in distinct colors. For improved visual clarity, the images have been cropped. Please kindly zoom in for a better view.

Table 8: Analysis experiments to evaluate parameters and performance of 3D UX-Net across various channel depths and the integration of uC on FLARE2021 and OIMHS datasets.

| Method | #Params | FLOPs | FLARE2021 | | | | | OIMHS | | | | |
| | | | mIoU | Dice | ASSD | HD95 | AdjRand | mIoU | Dice | ASSD | HD95 | AdjRand |
|---|---|---|---|---|---|---|---|---|---|---|---|---|
| 3D UX-Net (16) | 6.1M | 72.2G | 87.66±3.45 | 92.77±2.52 | 1.80±2.16 | 9.76±17.32 | 92.66±2.53 | 86.06±3.36 | 92.15±2.19 | 1.12±0.90 | 5.79±10.80 | 91.54±2.26 |
| 3D UX-Net (16) +uC | 6.7M | 97.1G | **88.09±2.36** | **93.12±1.59** | **1.05±0.61** | **2.99±1.28** | **93.01±1.61** | **89.60±3.19** | **94.28±2.00** | **0.36±0.40** | **2.47±4.95** | **93.84±2.07** |
| 3D UX-Net (24) | 13.4M | 160.2G | 88.19±2.54 | 93.24±1.64 | 1.63±2.11 | 7.46±11.84 | 93.12±1.67 | 87.86±3.15 | 93.27±1.95 | 0.64±0.67 | 3.68±6.63 | 92.74±2.03 |
| 3D UX-Net (24) +uC | 15.4M | 220.2G | **88.75±2.69** | **93.53±1.97** | **0.92±0.59** | **2.74±1.44** | **93.43±1.98** | **90.23±2.82** | **94.67±1.72** | **0.39±0.44** | **2.37±5.09** | **94.25±1.79** |
| 3D UX-Net (48) | 53.0M | 639.4G | 88.62±2.90 | 93.43±1.98 | 2.59±6.00 | 13.27±46.73 | 93.32±2.00 | 88.35±3.23 | 93.56±2.04 | 0.51±0.48 | 2.70±4.86 | 93.05±2.09 |
| 3D UX-Net (48) +uC | 60.6M | 867.91G | **89.62±2.31** | **94.11±1.52** | **0.76±0.35** | **2.53±1.34** | **94.01±1.55** | **90.38±3.47** | **94.73±2.15** | **0.34±0.42** | **2.39±5.00** | **94.33±2.22** |

is consistently observed across different layers of uC, highlighting its robust and efficient feature fusion capabilities. The focus of the proposed DFi thus shifts towards optimizing feature fusion efficiency rather than merely achieving performance improvements. Continuous improvement efforts will prioritize refining 2D and 3D feature integration while maintaining computational efficiency.

#### 4.4.4 Performance Comparison with Varying Channel Depths in 3D UX-Net

The low parameter performance ratio observed in 3D CNNs is largely due to their reliance on axially symmetric 3D convolutions, which extract spatial features well but struggle to capture critical axial-slice plane details. Introducing the uC structure significantly enhances the capability of 3D CNNs to extract axial-slice plane features. Can incorporating uC allow a reduction in channel count, thereby decreasing computational cost? To explore this, we conduct an analytical experiment utilizing 3D UX-Net as the backbone of FLARE2021 and OIMHS datasets to investigate the impact of channel depth and uC integration on segmentation performance. As illustrated in Table 8, achieving a 0.66% Dice improvement with the 3D UX-Net on FLARE2021 requires a tenfold model increase, highlighting the challenge of enhancing performance with 3D CNNs. The incorporation of uC mitigates the axial-slice plane performance drop-off, resulting in performance enhancements across 3D UX-Net with different channel depths. Notably, a 24-channel depth uC 3D UX-Net surpasses the original 48-channel depth 3D UX-Net on both datasets, with parameter count and FLOPs reduced to only 29% and 34.3%, respectively. This highlights uC's capability to achieve superior performance with fewer parameters, suggesting that substituting traditional skip connections with uC and reducing channel depth is a viable approach to achieve substantial reductions in parameter count and computational load while maintaining or even improving performance. The qualitative results are further illustrated in Fig. 4.

### 4.5 Ablation Studies

**Stage Selection for uC integration.** By replacing skip connections with uC at different stages, we validate the impact of introducing 2D axial-slice plane features at varying network layers on model performance. All models had an initial channel depth of 32.

Table 9: Ablation study of different stage selection for uC integration on the OIMHS dataset.

| Method | mIoU | Dice | ASSD | HD95 | AdjRand |
|---|---|---|---|---|---|
| 3D U-Net | 87.08±3.32 | 92.79±2.09 | 1.31±1.35 | 7.39±14.69 | 92.22±2.15 |
| +uC (stage 1) | 90.06±3.41 | 94.55±2.16 | 0.96±1.85 | 7.93±19.51 | 94.14±2.20 |
| +uC (stage 2) | 89.73±3.12 | 94.37±1.92 | 0.35±0.38 | 2.36±4.69 | 93.94±1.99 |
| +uC (stage 3) | 88.71±3.31 | 93.75±2.12 | 0.43±0.42 | 2.70±5.17 | 93.26±2.18 |
| +uC (stage 5) | 87.32±3.46 | 92.93±2.27 | 0.69±0.63 | 3.05±5.19 | 92.37±2.31 |
| +uC (stage 1, 2, 3) | 90.52±2.77 | 94.84±1.68 | 0.35±0.33 | **2.20±4.07** | 94.43±1.76 |
| +uC (stage 1, 2, 3) + DFI | **90.86±2.75** | **95.03±1.68** | **0.34±0.45** | 2.34±5.23 | **94.63±1.73** |

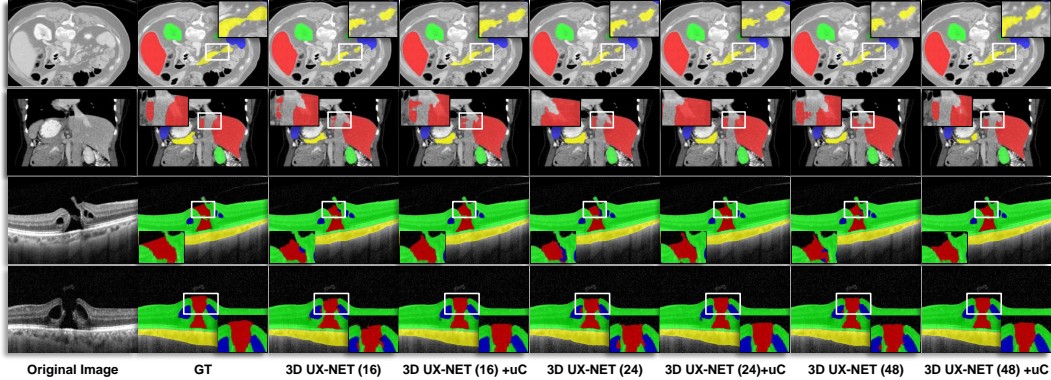

Figure 4: Qualitative results of the segmentation performance on 3D UX-Net and 3D UXNET+uC with various channel depths on the FLARE2021 and OIMHS datasets are presented. Segmentation results for different categories are represented in distinct colors. For improved visual clarity, the images have been cropped. Key regions of the qualitative results have been locally magnified for better viewing.

As shown in Table 9, supplementing initial feature information in shallower layers, closer to the original image, resulted in better performance improvements, with deeper layers showing diminishing returns. Hence, to balance parameter efficiency, we only replace skip connections at stages 1 to 3. Results indicate that using skip connections in the first three stages significantly enhances model performance, with the DFi module effectively integrating features extracted by 3D CNNs and uC.

**Channel Depth Configuration.** We experiment with channel depths of 16, 24, and 32 for the proposed uC 3DU-Net, all surpassing the original 32-channel U-Net, as shown in Table 10. Per-

Table 10: Ablation study of different Channel depths of uC 3DU-Net on the OIMHS dataset.

| Method | #Params | FLOPs | mIoU | Dice | ASSD | HD95 | AdjRand |
|---|---|---|---|---|---|---|---|
| 3D U-Net (32) | 4.81M | 135.9G | 87.08±3.32 | 92.79±2.09 | 1.31±1.35 | 7.39±14.69 | 92.22±2.15 |
| uC 3DU-Net (16) | 5.3M | 134.3G | 89.98±2.69 | 94.53±1.61 | 0.63±0.63 | 2.44±5.06 | 94.11±1.68 |
| uC 3DU-Net (24) | 21.7M | 286.4G | 90.67±2.95 | 94.91±1.84 | 0.34±0.46 | 2.35±5.22 | 94.53±1.89 |
| uC 3DU-Net (32) | 38.5M | 465.1G | **90.86±2.75** | **95.03±1.68** | **0.34±0.45** | **2.34±5.23** | **94.63±1.73** |

formance improved with increased channels, but efficiency dropped drastically at 32 channels compared to 24. Notably, the 16-channel uC 3DU-Net performed better than the 32-channel U-Net with similar parameter counts and FLOPs, demonstrating a 1.74% Dice score improvement and a 4.95 HD95 reduction. This underscores that efficiently capturing slice plane features with uC significantly enhances computational efficiency in 3D CNN architecture. Thus, replacing skip connections with uC and reducing channel depth is a straightforward strategy to achieve substantial parameter and computational load reductions while maintaining or enhancing performance.

# 5 Conclusion and future work

In this paper, we propose a U-shaped Connection (uC) for enhancing 3D CNN-based medical image segmentation architecture. This approach is specifically designed to address the inherent axial-slice plane performance drop-off in 3D CNNs, characterized by their inefficiency in extracting high-density axial-slice plane features, which are crucial for accurate 3D medical image segmentation. By replacing traditional skip connections in 3D U-Net with uC, we further develop the uC 3DU-Net, capitalizing on more efficient feature extraction capabilities of both 3D sequential spatial features and 2D axial-slice plane features, reaching the best segmentation accuracy and computational efficiency ratio among all previous SoTA methods. Empirical evaluations on diverse datasets demonstrate that uC 3DU-Net consistently outperforms previous SoTA 3D medical segmentation methods, with notably reduced parameters and FLOPs. This underscores the transformative potential of the uC structure in revolutionizing medical volumetric segmentation by breaking through the intrinsic limitations of 3D convolutions. Future research will extend the application of the uC structure to a broader range of volumetric segmentation tasks, with the aim of continually advancing the performance and efficiency of 3D image segmentation models.

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

# Appendix

We first introduce the details of the datasets and implementation in Sec. A, followed by the supplementary experiments and analysis in Sec. B. Furthermore, the experimental results are detailed in Sec. C. Lastly, additional qualitative results are provided in Sec. D.

# A  Details of Datasets and Implementation

## A.1  Public datasets details

Here, we provide detailed information on the five public datasets used for experiments in Table 11.

Table 11: Detailed information of five publicly available Datasets: FLARE2021, OIMHS, FeTA02021, AbdomenCT-1K, and BTCV.

| Datasets | FLARE 2021 | OIMHS |
|---|---|---|
| Imaging Modality | Multi-Contrast CT | OCT |
| Anatomical Region | Abdomen | Eye |
| Sample Size | 361 | 125 |
| Dimensions | $512 \times 512 \times \{37 - 751\}$ | $512 \times 512 \times \{19 - 79\}$ |
| Resolution | $\{0.61 - 0.98\}mm \times \{0.61 - 0.98\}mm \times \{0.50 - 7.50\}mm$ | $\{10.7 - 14.0\}um \times \{10.7 - 14.0\}um \times \{7.0 - 40.0\}um$ |
| Anatomical Label | Spleen, Kidney, Liver, Pancreas | Retinal, Choroid, Macular hole, Macular edema |

| Datasets | FeTA2021 | AbdomenCT-1K |
|---|---|---|
| Imaging Modality | 1.5T & 3T MRI | Multi-Contrast CT |
| Anatomical Region | Infant Brain | Abdomen |
| Sample Size | 80 | 361 |
| Dimensions | $256 \times 256 \times 256$ | $\{512 - 796\} \times \{79 - 512\} \times \{31 - 1026\}$ |
| Resolution | $\{0.43 - 1.00\}mm \times \{0.43 - 1.00\}mm \times \{0.43 - 1.00\}mm$ | $\{0.45 - 1.04\}mm \times \{0.45 - 3.00\}mm \times \{0.45 - 8.00\}mm$ |
| Anatomical Label | External cerebrospinal fluid, Grey matter, White matter, Ventriculus, Cerebellum, Deep grey matter, Brainstem | Spleen, Kidney, Liver, Pancreas |

| Datasets | BTCV |
|---|---|
| Imaging Modality | Multi-Contrast CT |
| Anatomical Region | Abdomen |
| Sample Size | 30 |
| Dimensions | $512 \times 512 \times \{85 - 198\}$ |
| Resolution | $\{0.59 - 0.98\}mm \times \{0.59 - 0.98\}mm \times \{2.50 - 5.00\}mm$ |
| Anatomical Label | Spleen, Kidney right, Kidney left, Gallbladder Esophagus, Liver, Stomach, Aorta Inferior vena cava, portal vein and splenic vein, pancreas adrenal gland right, adrenal gland left |

## A.2  Data Preprocessing & Implementation Detail

In the 4.2, we provide a detailed description of our data preprocessing process and offer more comprehensive hyperparameter settings in Table 12. For the BTCV dataset, we specifically incorporate resampling to specific pixel spacing(1.5, 1.5, 2.0) and random intensity shift as additional data augmentation strategies.

Table 12: Detail hyperparameters of training scenarios on five public datasets.

| Training Steps | 80000 |
|---|---|
| Batch Size | 2 |
| AdamW$_\epsilon$ | $1e-8$ |
| AdamW$_\beta$ | (0.9, 0.999) |
| Peak Learning Rate | $1e-4$ |
| Learning Rate Scheduler | ReduceLROnPlateau |
| Factor & Patience | 0.9, 5 |
| Data Augmentation | Flip, Rotation, Scaling
Intensity Shift, 3D elastic transformation |
| Cropped Foreground | ✓ |
| Rotation Degree | $(-30°, +30°)$ |
| Scaling Factor | 0.1 |
| Intensity Offset | 0.1 |

# B  Supplementary experiments and analysis

## B.1  Benchmarking results on BTCV

We present comparative experiments of the proposed uC 3D-UNet against previous classic segmentation methods on the BTCV dataset, as shown in Table 13. Given the smaller dataset size, CNN-based models outperformed Transformer-based models. The results indicate that uC 3D-UNet maintains outstanding performance on the challenging 13 classes BTCV segmentation dataset.

Table 13: Comparative experimental results of uC 3DU-Net and 4 previous methods on the BTCV Standard dataset. The best values for each metric are highlighted in bold.

| Methods | #Params | FLOPs | mIoU | Dice | ASSD | HD95 | AdjRand |
|---|---|---|---|---|---|---|---|
| 3D U-Net | 4.8M | 135.9G | 72.25±3.30 | 82.24±2.99 | 1.06±0.40 | 4.22±1.35 | 82.19±3.00 |
| TransBTS | 31.6M | 110.4G | 67.85±3.55 | 78.38±3.50 | 2.01±1.03 | 8.70±4.60 | 78.31±3.51 |
| UNETR | 92.8M | 82.6G | 68.99±2.52 | 79.82±2.51 | 1.39±0.54 | 7.38±4.90 | 79.76±2.51 |
| 3D UX-Net | 53.0M | 639.4G | 72.27±3.26 | 82.31±3.06 | 1.37±0.45 | 4.70±1.49 | 82.25±3.06 |
| uC 3DU-Net | 21.7M | 286.4G | **72.99±3.40** | **82.74±3.18** | **0.97±0.21** | **3.63±0.80** | **82.69±3.19** |

## B.2  Additional comparative experiments

Here, we add two recently proposed models as baselines for comparative experiments across four publicly available datasets: FLARE2021, OIMHS, FeTA2021, and AbdomenCT-1K. The experimental results in Table 14 demonstrate that our proposed uC 3D-UNet maintains superior performance compared to the latest models.

Table 14: Comparative experimental results of the proposed uC 3D-UNet against two added baselines on the FLARE2021, FeTA2021, OIMHS, and AbdomenCT-1K datasets.

| Datasets | #Params | FLOPs | Methods | mIoU | Dice | ASSD | HD95 | AdjRand |
|---|---|---|---|---|---|---|---|---|
| FLARE2021 | UNETR++ [30] | 68.59M | 19.75G | 83.98±4.45 | 89.92±3.73 | 1.10±0.51 | 4.14±1.93 | 89.79±3.75 |
|  | D-LKA Net [62] | 34.64M | 55.32G | 88.80±2.58 | 93.56±1.82 | 0.71±0.28 | 2.53±1.05 | 93.46±1.83 |
|  | uC 3DU-Net | 21.7M | 286.4G | **89.36±2.26** | **93.98±1.46** | **0.68±0.30** | **2.35±0.99** | **93.88±1.49** |
| FeTA2021 | UNETR++ [30] | 68.59M | 19.75G | 72.90±2.60 | 84.04±1.82 | 1.19±0.11 | 3.37±0.85 | 83.56±1.75 |
|  | D-LKA Net [62] | 34.64M | 55.32G | 73.35±3.83 | 84.28±2.64 | 1.19±0.16 | 3.19±0.65 | 83.82±2.56 |
|  | uC 3DU-Net | 21.7M | 286.4G | **78.66±3.05** | **87.88±1.98** | **0.91±0.09** | **2.34±0.27** | **87.48±1.91** |
| OIMHS | UNETR++ [30] | 68.59M | 19.75G | 79.27±7.49 | 87.08±6.54 | 0.99±0.65 | 4.83±2.49 | 86.26±6.54 |
|  | D-LKA Net [62] | 34.64M | 55.32G | 86.55±4.11 | 92.39±2.83 | 0.44±0.27 | **2.28±1.12** | 91.84±2.87 |
|  | uC 3DU-Net | 21.7M | 286.4G | **90.67±2.95** | **94.91±1.84** | **0.34±0.46** | 2.35±5.22 | **94.53±1.89** |
| AbdomenCT-1K | UNETR++ [30] | 68.59M | 19.75G | 81.11±15.16 | 87.20±15.19 | 2.69±4.84 | 9.70±12.43 | 87.04±15.24 |
|  | D-LKA Net [62] | 34.64M | 55.32G | 87.84±3.35 | 93.08±2.13 | 1.49±0.94 | **7.16±9.08** | 92.95±2.18 |
|  | uC 3DU-Net | 21.7M | 286.4G | **88.29±4.04** | **93.35±2.60** | **1.48±1.09** | 8.53±10.74 | **93.22±2.66** |

## B.3  Dimensional Slicing Analysis

To validate the effectiveness of the proposed uC in enhancing axial-slice plane feature extraction, we conduct experiments with 3D U-Net + 2D uC by slicing input tensors along different dimensions. As shown in Table 15, although the 2D slicing augmentation with uC consistently enhances performance across all planes, slicing along the time-axial slice plane dimension yields the best results. This result reflects the denser information available in the axial-slice plane of 3D medical images, which improves performance when using the proposed uC method. Furthermore, these findings support why 2D uC achieves superior parameter efficiency in 3D image segmentation tasks compared to purely 3D convolutions.

Table 15: The results of applying 2D uC on slices along the W, H, and D dimensions on the OIMHS dataset. The best values for each metric are highlighted in bold.

| Method | Average | | | | | Macular Hole | | | | |
|---|---|---|---|---|---|---|---|---|---|---|
| | mIOU | Dice | ASSD | HD95 | AdjRand | mIOU | Dice | ASSD | HD95 | AdjRand |
| 3D U-Net(32) | 87.08±3.32 | 92.79±2.09 | 1.31±1.35 | 7.39±14.69 | 92.22±2.15 | 77.89±6.74 | 87.42±4.28 | 3.94±5.25 | 24.03±58.65 | 87.28±4.30 |
| +2D uC(W) | 88.01±3.12 | 93.35±1.96 | 0.86±0.66 | 2.89±5.53 | 92.84±2.02 | 80.07±5.99 | 88.81±3.80 | 2.39±2.31 | 6.63±21.77 | 88.68±3.81 |
| +2D uC(H) | 87.74±3.74 | 93.16±2.46 | 0.48±0.47 | 2.91±5.03 | 92.63±2.50 | 79.96±7.08 | 88.69±4.59 | 0.88±1.68 | 6.28±19.91 | 88.56±4.60 |
| +2D uC(D) | **90.67±2.95** | **94.91±1.84** | **0.34±0.46** | **2.35±5.22** | **94.53±1.89** | **83.57±5.21** | **90.96±3.13** | **0.80±1.69** | **5.80±20.64** | **90.86±3.17** |

| Retinal | | | | | Macular Edema | | | | | Choroid | | | | |
|---|---|---|---|---|---|---|---|---|---|---|---|---|---|---|
| mIOU | Dice | ASSD | HD95 | AdjRand | mIOU | Dice | ASSD | HD95 | AdjRand | mIOU | Dice | ASSD | HD95 | AdjRand |
| 97.68±1.52 | 98.82±0.80 | 0.17±0.13 | 1.15±0.51 | 98.05±1.32 | 80.39±8.28 | 88.89±5.38 | 0.54±1.06 | 1.52±1.44 | 88.75±5.38 | 92.36±2.43 | 96.01±1.32 | 0.58±0.29 | 2.85±1.97 | 94.83±1.72 |
| 97.75±1.55 | 98.86±0.81 | 0.16±0.13 | 1.15±0.50 | 98.12±1.33 | 80.86±8.06 | 89.18±5.20 | 0.41±0.74 | 1.45±1.24 | 89.04±5.21 | 93.37±2.07 | 96.56±1.11 | 0.47±0.30 | 2.34±1.96 | 95.53±1.53 |
| 97.72±1.44 | 98.84±0.75 | 0.16±0.12 | 1.12±0.42 | 98.09±1.24 | 80.38±8.81 | 88.84±5.93 | 0.35±0.41 | 1.41±0.88 | 88.70±5.93 | 92.88±2.30 | 96.30±1.25 | 0.54±0.34 | 2.83±2.33 | 95.17±1.68 |
| **97.99±1.51** | **98.98±0.79** | **0.14±0.13** | **1.12±0.43** | **98.32±1.30** | **85.16±8.18** | **91.76±5.26** | **0.21±0.22** | **1.25±1.13** | **91.65±5.26** | **95.95±1.18** | **97.93±0.62** | **0.23±0.10** | **1.22±0.41** | **97.30±0.83** |

## B.4 Visualization of the Impact of Convolution Dimensionality on uC Performance and Parameters

Here, we present the visualization results from Sec. 4.4.1, as shown in Fig. 5.

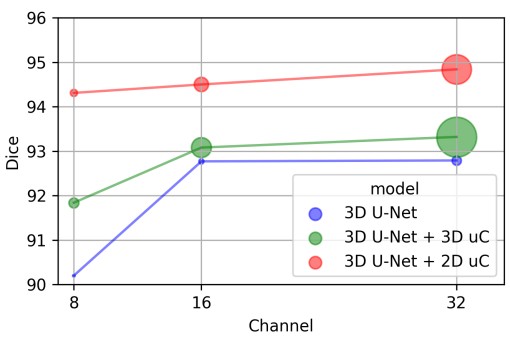
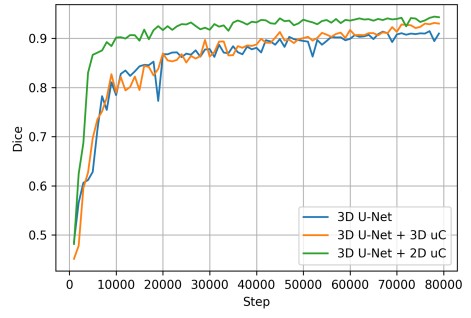

(a) Performance comparison of 3D U-Net, 3D U-Net + 3D uC, and 3D U-Net + 2D uC across varying channel depths, with circle size representing parameter count. The horizontal axis represents channel depth numbers.

(b) Validation curve showing Dice scores on the OIMHS dataset, with the channel depth for all three models(3D U-Net, 3D U-Net + 3D uC, and 3D U-Net + 2D uC) set to 32. The horizontal axis represents the number of training iterations.

Figure 5: Visualization of the impact of 2D and 3D Conv on uC. (a) depicts the relationship between performance and channel depth. (b) shows the validation curve indicating the training efficiency.

## C Details of experimental results

In this section, we present the details of our experimental results. For comparative experiments, as the baselines for FLARE2021 and FeTA2021 utilize data from [56], we provide detailed experimental results for all categories in the OIMHS and AbdomenCT-1K datasets (with part of OIMHS comparative data comes from [25], which are not included here), as shown in Tables 16 and 17. Tables 18, 19, 20, 21, and 22 present detailed data for all categories in the analytical experiments, while Tables 23 and 24 display the detailed data for all categories in the ablation studies. The comprehensive results across all categories further demonstrate that integrating the uC module significantly enhances the backbone models' ability to extract 2D plane information from axial slices, while retaining the extensive volumetric feature extraction capabilities of 3D convolutions. This integration effectively addresses the challenge of anisotropic medical imaging, achieving superior performance with reduced computational complexity and fewer parameters compared to previous models. As a result, the proposed approach achieves an optimal parameter-to-performance ratio.

Table 16: Detailed results of uC 3DU-Net and 3 previous methods on the OIMHS dataset. The best values for each metric are highlighted in bold.

| Method | #Params | FLOPs | Average IoU | Dice | VOE | HD95 | AdjRand | Macular Hole IoU | Dice | VOE | HD95 | AdjRand |
|---|---|---|---|---|---|---|---|---|---|---|---|---|
| nnFormer | 149.3M | 240.2G | 72.16±7.91 | 81.60±7.41 | 27.84±25.07 | 23.49±7.91 | 80.36±7.73 | 50.54±18.06 | 64.92±18.08 | 49.46±78.16 | 65.84±18.06 | 64.63±18.07 |
| TransBTS | 31.6M | 110.4G | 74.80±7.31 | 83.08±6.55 | 25.20±23.82 | 31.43±7.31 | 82.05±6.85 | 47.24±17.58 | 60.85±17.85 | 52.76±88.66 | 107.45±17.58 | 60.60±17.94 |
| UNETR | 92.8M | 82.6G | 80.52±6.68 | 88.11±5.41 | 19.48±30.31 | 30.07±6.68 | 87.21±5.56 | 64.53±15.37 | 77.17±12.99 | 35.47±99.84 | 93.03±15.37 | 76.96±12.99 |
| uC 3DU-Net | 21.7M | 286.43G | **89.48±3.56** | **94.13±2.56** | **10.52±5.61** | **2.98±3.56** | **93.62±2.66** | **80.89±7.40** | **89.22±4.84** | **19.11±21.67** | **7.41±7.40** | **89.09±4.86** |

| Retinal IoU | Dice | VOE | HD95 | AdjRand | Macular Edema IoU | Dice | VOE | HD95 | AdjRand | Choroid IoU | Dice | VOE | HD95 | AdjRand |
|---|---|---|---|---|---|---|---|---|---|---|---|---|---|---|
| 95.69±3.07 | 97.76±1.70 | 4.31±8.94 | 3.52±3.07 | 96.30±2.84 | 58.84±12.87 | 72.94±12.29 | 41.16±30.74 | 15.66±12.87 | 72.58±12.33 | 83.56±8.03 | 90.79±5.26 | 16.44±8.92 | 8.95±8.03 | 87.93±6.69 |
| 96.66±2.43 | 98.30±1.30 | 3.34±1.34 | 1.40±2.43 | 97.24±2.14 | 68.78±12.50 | 80.52±10.60 | 31.22±16.76 | 6.18±12.50 | 80.25±10.69 | 87.15±5.95 | 92.99±3.59 | 12.85±15.82 | 10.48±5.95 | 90.70±5.06 |
| 96.96±1.87 | 98.45±0.98 | 3.04±1.91 | 1.53±1.87 | 97.43±1.63 | 73.01±12.36 | 83.59±10.42 | 26.99±39.50 | 20.31±12.36 | 83.36±10.43 | 87.56±5.94 | 93.24±3.58 | 12.44±5.02 | 5.41±5.94 | 91.09±4.72 |
| **97.96±1.22** | **98.97±0.63** | **2.04±0.28** | **1.06±1.22** | **98.29±1.07** | **85.12±8.71** | **91.53±6.80** | **14.88±1.83** | **1.35±8.71** | **91.39±6.85** | **93.92±3.37** | **96.83±1.87** | **6.08±2.11** | **2.11±3.37** | **95.72±2.73** |

Table 17: Detailed results of uC 3DU-Net and 6 previous methods on the AbdomenCT-1K dataset. The best values for each metric are highlighted in bold.

| Method | #Params | FLOPs | Average IoU | Dice | ASSD | HD95 | AdjRand | Liver IoU | Dice | ASSD | HD95 | AdjRand |
|---|---|---|---|---|---|---|---|---|---|---|---|---|
| 3D U-Net | 4.81M | 135.9G | 86.69±4.30 | 92.28±2.93 | 2.31±2.06 | 12.68±18.54 | 92.14±2.98 | 93.97±4.06 | 96.85±2.21 | 3.57±3.66 | 22.45±38.69 | 96.53±2.38 |
| TransBTS | 31.6M | 110.4G | 70.06±9.11 | 79.64±8.12 | 4.88±1.59 | 35.75±10.00 | 79.35±8.19 | 87.79±6.29 | 93.37±3.85 | 10.65±3.67 | 106.69±28.27 | 92.70±4.14 |
| UNETR | 92.8M | 82.6G | 84.17±5.12 | 90.42±3.75 | 2.36±1.68 | 13.41±16.77 | 90.27±3.81 | 93.34±3.81 | 96.51±2.06 | 3.98±3.33 | 25.80±39.37 | 96.17±2.22 |
| nnFormer | 149.3M | 240.2G | 80.69±7.76 | 87.56±6.36 | 2.01±1.80 | 9.75±9.95 | 87.38±6.41 | 92.52±4.81 | 96.05±2.70 | 1.14±1.16 | **5.35±8.41** | 95.66±2.90 |
| Swin UNETR | 62.2M | 328.4G | 86.76±4.94 | 92.36±3.20 | 2.67±1.81 | 14.87±18.78 | 92.22±3.26 | 94.15±3.84 | 96.95±2.08 | 4.11±2.83 | 21.27±34.64 | 96.64±2.25 |
| 3D UX-Net | 53.0M | 639.4G | 86.56±4.76 | 92.21±3.18 | 2.60±1.98 | 15.46±19.75 | 92.07±3.24 | 94.34±3.73 | 97.05±2.01 | 3.84±3.57 | 27.27±49.39 | 96.75±2.19 |
| uC 3DU-Net | 21.7M | 286.4G | **88.29±4.04** | **93.35±2.60** | **1.48±1.09** | **8.53±10.74** | **93.22±2.66** | **95.07±3.53** | **97.44±1.90** | **1.71±1.70** | 7.92±19.60 | **97.18±2.07** |

| Kidney IoU | Dice | ASSD | HD95 | AdjRand | Spleen IoU | Dice | ASSD | HD95 | AdjRand | Pancreas IoU | Dice | ASSD | HD95 | AdjRand |
|---|---|---|---|---|---|---|---|---|---|---|---|---|---|---|
| 91.41±4.17 | 95.57±2.31 | 1.12±1.06 | 4.75±7.98 | 95.47±2.36 | 93.74±3.80 | 96.73±2.08 | 2.01±3.07 | 10.51±27.27 | 96.68±2.10 | 67.44±11.37 | 79.97±8.93 | 2.55±3.35 | 13.00±33.34 | 79.88±8.97 |
| 90.76±4.91 | 95.09±2.77 | **0.76±0.56** | **2.51±1.76** | 94.97±2.81 | 42.68±19.15 | 57.32±19.45 | 4.94±2.91 | 16.42±11.67 | 57.05±19.46 | 59.02±16.40 | 72.76±14.88 | 3.16±3.16 | 17.37±30.75 | 72.65±14.92 |
| 90.11±5.82 | 94.70±3.34 | 1.55±3.38 | 8.62±24.73 | 94.58±3.40 | 93.37±4.06 | 96.53±2.26 | **0.75±1.03** | **1.83±1.68** | 96.48±2.28 | 59.85±13.22 | 73.96±11.67 | 3.14±2.76 | 17.40±31.12 | 73.84±11.72 |
| 88.53±6.20 | 93.79±3.77 | 1.19±1.23 | 6.66±16.43 | 93.65±3.83 | 89.71±13.66 | 93.80±11.31 | 0.90±1.78 | 3.94±8.91 | 93.73±11.35 | 52.01±17.19 | 66.60±16.91 | 4.80±6.40 | 23.06±34.00 | 66.48±16.94 |
| 91.75±4.87 | 95.63±2.73 | 1.29±1.07 | 5.53±2.77 | 95.54±2.68 | 92.63±6.25 | 96.06±3.60 | 3.22±5.07 | 23.96±44.92 | 96.01±3.64 | 68.49±10.65 | 80.81±8.07 | 2.04±1.65 | 9.18±11.35 | 80.72±8.12 |
| 91.51±4.70 | 95.50±2.62 | 0.95±0.81 | 3.66±4.82 | 95.40±2.68 | 92.33±6.04 | 95.91±3.41 | 3.88±5.84 | 25.07±45.80 | 95.85±3.45 | 68.07±11.80 | 80.38±9.18 | 1.74±0.98 | 5.84±5.12 | 80.29±9.22 |
| **92.21±4.37** | **95.89±2.44** | 0.95±1.07 | 4.58±10.93 | **95.79±2.49** | **94.14±4.77** | **96.92±2.69** | 1.88±3.36 | 16.88±39.37 | **96.88±2.72** | **71.72±9.72** | **83.14±7.21** | **1.39±0.68** | **4.73±3.77** | **83.05±7.25** |

Table 18: The detailed results of 3D U-Net, 3D U-Net + 3D uC, and 3D U-Net + 2D uC across different channel depth numbers on the OIMHS dataset. The best values for each metric are highlighted in bold.

| Method | #Params | FLOPs | Average mIoU | Dice | ASSD | HD95 | AdjRand | Macular Hole mIoU | Dice | ASSD | HD95 | AdjRand |
|---|---|---|---|---|---|---|---|---|---|---|---|---|
| 3D U-Net(8) | 0.37M | 16.50G | 83.17±4.51 | 90.20±3.20 | 2.88±2.16 | 16.92±21.89 | 89.51±3.23 | 71.22±8.29 | 82.91±5.84 | 9.48±8.42 | 60.45±87.91 | 82.73±5.83 |
| ±3D uC | 5.56M | 46.80G | 85.58±3.60 | 91.84±2.35 | 2.25±1.35 | 8.64±14.07 | 91.19±2.40 | 74.04±5.84 | 85.01±4.69 | 5.01±4.69 | 27.70±55.93 | 84.79±3.86 |
| ±2D uC | 2.85M | 42.54G | 89.59±2.61 | 94.31±1.62 | 0.52±0.48 | 2.57±4.74 | 93.84±1.65 | 82.09±4.74 | 90.09±2.91 | 1.13±1.70 | 5.99±19.05 | 89.97±2.94 |
| 3D U-Net(16) | 1.47M | 65.09G | 87.08±3.43 | 92.77±2.24 | 0.92±0.92 | 4.57±8.92 | 92.20±2.29 | 79.15±6.19 | 88.23±3.96 | 1.57±1.85 | 6.54±19.86 | 88.10±3.97 |
| ±3D uC | 22.21M | 186.27G | 87.50±3.34 | 93.08±2.11 | 4.61±1.25 | 43.45±19.56 | 92.41±2.15 | 79.53±6.99 | 88.42±4.52 | 1.23±1.93 | 7.36±22.24 | 88.29±4.53 |
| ±2D uC | 11.38M | 169.20G | 89.98±3.43 | 94.50±2.23 | 0.39±0.42 | 2.56±4.96 | 94.06±2.26 | 81.69±7.32 | 89.72±4.92 | 0.80±1.57 | 6.11±19.75 | 89.61±4.94 |
| 3D U-Net(32) | 4.81M | 135.90G | 87.08±3.32 | 92.79±2.09 | 1.31±1.35 | 7.39±14.69 | 92.22±2.15 | 77.89±6.74 | 87.42±4.28 | 3.94±5.25 | 24.03±58.65 | 87.28±4.30 |
| ±3D uC | 88.67M | 596.06G | 87.93±3.33 | 93.32±2.09 | 3.09±1.31 | 14.13±16.90 | 92.73±2.12 | 79.63±6.31 | 88.52±4.02 | 2.13±3.46 | 14.86±43.19 | 88.39±4.04 |
| ±2D uC | 45.33M | 527.75G | 90.52±2.77 | 94.84±1.68 | 0.35±0.33 | 2.20±4.07 | 94.43±1.76 | 82.66±5.63 | 90.40±3.48 | 0.76±1.16 | 5.11±16.15 | 90.29±3.51 |

| Retinal mIoU | Dice | ASSD | HD95 | AdjRand | Macular Edema mIoU | Dice | ASSD | HD95 | AdjRand | Choroid mIoU | Dice | ASSD | HD95 | AdjRand |
|---|---|---|---|---|---|---|---|---|---|---|---|---|---|---|
| 97.23±1.71 | 98.59±0.90 | 0.34±0.51 | 1.25±0.67 | 97.67±1.50 | 73.46±11.47 | 84.14±8.41 | 0.93±1.54 | 2.22±2.19 | 83.95±8.40 | 90.78±2.76 | 95.15±1.54 | 0.77±0.39 | 3.75±2.44 | 93.68±1.94 |
| 97.02±1.54 | 98.48±0.81 | 1.75±1.05 | 1.31±0.64 | 97.50±1.33 | 79.22±9.15 | 88.09±6.14 | 1.21±1.59 | 2.17±2.92 | 87.93±6.14 | 92.02±2.57 | 95.83±1.41 | 1.04±0.47 | 3.38±2.32 | 94.55±1.89 |
| 97.85±1.40 | 98.91±0.73 | 0.15±0.12 | 1.12±0.42 | 98.20±1.21 | 84.49±7.46 | 91.40±4.72 | 0.28±0.30 | 1.24±0.88 | 91.29±4.72 | 93.91±1.85 | 96.85±0.99 | 0.53±0.53 | 1.94±0.92 | 95.89±1.27 |
| 97.61±1.55 | 98.78±0.81 | 0.18±0.14 | 1.15±0.52 | 97.99±1.34 | 79.26±9.11 | 88.12±6.20 | 1.03±2.94 | 7.53±30.75 | 87.96±6.20 | 92.28±2.35 | 95.97±1.29 | 0.92±1.45 | 3.07±2.28 | 94.74±1.67 |
| 97.77±1.62 | 98.87±0.84 | 0.15±0.52 | 1.15±1.39 | 98.13±1.39 | 83.54±8.32 | 90.79±5.33 | 0.34±0.34 | 1.31±0.63 | 90.67±5.34 | 89.16±3.30 | 94.23±1.85 | 16.61±4.80 | 164.00±75.55 | 92.55±2.14 |
| 97.96±1.47 | 98.96±0.76 | 0.14±0.12 | 1.13±0.45 | 98.29±1.27 | 85.72±7.31 | 92.13±4.57 | 0.25±0.29 | 1.02±0.59 | 92.02±4.58 | 94.57±1.93 | 97.20±1.03 | 0.36±0.23 | 1.97±1.57 | 96.32±1.46 |
| 97.68±1.52 | 98.82±0.80 | 0.17±0.13 | 1.15±0.51 | 98.05±1.32 | 80.39±8.28 | 88.89±5.38 | 0.54±1.06 | 1.52±1.44 | 88.75±5.38 | 92.36±2.43 | 96.01±1.32 | 0.58±0.29 | 2.85±1.97 | 94.83±1.72 |
| 97.86±1.54 | 98.91±0.81 | 0.21±0.14 | 1.15±0.52 | 98.20±1.53 | 83.22±8.51 | 90.58±5.49 | 0.92±2.29 | 6.62±26.38 | 90.46±5.50 | 91.01±2.74 | 95.27±1.51 | 9.11±2.76 | 33.90±45.50 | 93.86±1.89 |
| 98.03±1.48 | 99.00±0.77 | 0.13±0.12 | 1.12±0.43 | 98.35±1.27 | 85.94±6.96 | 92.29±4.28 | 0.22±0.22 | 1.02±0.51 | 92.18±4.28 | 95.44±1.60 | 97.66±0.84 | 0.29±0.19 | 1.55±1.15 | 96.92±1.30 |

Table 19: The detailed results of the analysis experiments evaluating the effectiveness and robustness of the uC across different backbones on the FLARE2021 dataset.

| Method | #Params | FLOPs | Average IoU | Dice | ASSD | HD95 | AdjRand | Liver IoU | Dice | ASSD | HD95 | AdjRand |
|---|---|---|---|---|---|---|---|---|---|---|---|---|
| SegResNet | 18.8M | 244.4G | 88.38±2.07 | 93.39±1.35 | 1.31±1.29 | 6.37±10.45 | 93.28±1.37 | 96.48±1.14 | 98.20±0.59 | 0.55±0.33 | **1.48±0.45** | 98.00±0.70 |
| **SegResNet +uC** | 20.5M | 426.7G | 89.42±1.71 | 94.04±1.05 | 0.80±0.45 | 2.35±1.03 | 93.94±1.08 | 96.61±1.31 | 98.27±0.68 | 0.50±0.32 | 1.55±0.76 | 98.07±0.79 |
| TransBTS | 31.6M | 110.4G | 87.63±2.74 | 92.84±1.86 | 1.04±0.64 | 3.54±2.64 | 92.73±1.88 | 96.27±1.11 | 98.09±0.57 | 0.52±0.16 | 1.58±0.49 | 97.88±0.67 |
| **TransBTS +uC** | 36.6M | 176.4G | 89.46±2.03 | 94.05±1.32 | 0.78±0.30 | 2.51±1.17 | 93.95±1.34 | 96.45±1.23 | 98.19±0.64 | 0.50±0.22 | 1.53±0.44 | 97.98±0.73 |
| Swin UNETR | 62.2M | 328.4G | 88.28±2.69 | 93.23±1.83 | 0.95±0.41 | 3.25±2.18 | 93.13±1.85 | 96.42±1.24 | 98.17±0.64 | 0.53±0.29 | 1.63±0.56 | 97.97±0.74 |
| **Swin UNETR +uC** | 75.4M | 612.2G | 89.07±2.60 | 93.72±1.83 | 0.76±0.41 | 2.51±1.33 | 93.62±1.85 | 96.58±1.17 | 98.26±0.60 | 0.50±0.27 | 1.54±0.51 | 98.05±0.72 |
| 3D UX-Net | 53.0M | 639.4G | 88.62±2.90 | 93.43±1.98 | 2.59±6.00 | 13.27±46.73 | 93.32±2.00 | 96.48±1.33 | 98.20±0.69 | 1.49±2.83 | 9.76±35.47 | 98.00±0.79 |
| **3D UX-Net +uC** | 60.6M | 867.9G | 89.62±2.31 | 94.11±1.52 | 0.76±0.35 | 2.53±1.34 | 94.01±1.55 | 96.73±1.24 | 98.34±0.64 | 0.46±0.25 | 1.47±0.46 | 98.14±0.76 |

| Kidney IoU | Dice | ASSD | HD95 | AdjRand | Spleen IoU | Dice | ASSD | HD95 | AdjRand | Pancreas IoU | Dice | ASSD | HD95 | AdjRand |
|---|---|---|---|---|---|---|---|---|---|---|---|---|---|---|
| 92.30±2.58 | 95.98±1.41 | 0.82±0.46 | 2.76±2.00 | 95.88±1.45 | 95.49±1.51 | 97.69±0.79 | **0.63±0.78** | 1.18±0.38 | 97.65±0.80 | 69.26±6.21 | 81.68±4.38 | 3.25±4.05 | 20.07±39.91 | 81.59±4.39 |
| 94.02±2.68 | 96.90±1.44 | 0.56±0.43 | 2.14±2.59 | 96.82±1.48 | 95.69±1.63 | 97.79±0.86 | 0.71±0.88 | 1.17±0.33 | 97.75±0.87 | 71.36±4.56 | 83.21±3.13 | 1.45±0.64 | 4.53±1.75 | 83.12±3.15 |
| 92.15±2.96 | 95.89±1.62 | 1.15±1.09 | 3.87±6.12 | 95.79±1.66 | 95.39±1.58 | 97.63±0.83 | 0.48±0.33 | 1.20±0.37 | 97.59±0.84 | 66.70±8.00 | 79.76±5.91 | 2.01±1.54 | 7.52±6.99 | 79.66±5.93 |
| 93.50±2.57 | 96.62±1.39 | 0.62±0.42 | 2.18±2.27 | 96.54±1.42 | 96.13±1.43 | 98.02±0.75 | 0.56±0.52 | 1.13±0.23 | 97.99±0.76 | 71.78±7.02 | 83.38±4.85 | 1.44±0.62 | 5.19±3.48 | 83.30±4.87 |
| 93.03±3.34 | 96.36±1.80 | 1.01±0.83 | 4.39±8.93 | 96.27±1.86 | 95.81±1.50 | 97.85±0.79 | 0.68±1.04 | 1.50±0.75 | 97.81±0.80 | 69.33±8.49 | 80.55±6.25 | **1.64±0.71** | 5.72±2.69 | 80.45±6.25 |
| 94.13±3.04 | 96.95±1.63 | 0.54±0.46 | 2.31±2.67 | 96.88±1.67 | 96.30±1.44 | 98.11±0.75 | 0.32±0.18 | 1.11±0.19 | 98.08±0.76 | 69.29±8.14 | 81.57±6.28 | 1.68±1.03 | 5.27±2.86 | 81.48±6.29 |
| 93.87±3.01 | 96.81±1.62 | **0.62±0.46** | 2.27±2.56 | 96.74±1.66 | 95.73±1.84 | 97.81±0.97 | 3.28±9.21 | 20.42±83.86 | 97.77±0.98 | 68.40±9.31 | 80.88±6.78 | 4.98±12.51 | 20.64±67.90 | 80.79±6.81 |
| 94.05±2.90 | 96.91±1.56 | 0.63±0.46 | 2.29±2.52 | 96.84±1.60 | 96.34±1.33 | 98.13±0.69 | 0.51±0.48 | 1.13±0.23 | 98.10±0.70 | 71.35±7.46 | 83.06±5.27 | 1.45±0.63 | 5.25±4.41 | 82.98±5.29 |

Table 20: The detailed results of the analysis experiments on DFI are presented to demonstrate its efficiency and effectiveness on the OIMHS dataset. The best values for each metric are highlighted in bold.

| Method | #Params | FLOPs | mIOU | Dice | Average ASSD | HD95 | AdjRand | Macular Hole mIOU | Dice | ASSD | HD95 | AdjRand |
|---|---|---|---|---|---|---|---|---|---|---|---|---|
| uC(stage1) | 30.38M | 439.87G | 90.06±3.41 | 94.55±2.16 | 0.96±1.85 | 7.93±19.51 | 94.14±2.20 | 82.49±6.14 | 90.28±3.85 | 0.72±1.68 | 5.89±20.82 | 90.17±3.88 |
| +DFi | 23.58M | 377.19G | 90.42±2.59 | 94.79±1.56 | 0.68±1.19 | 4.45±11.03 | 94.38±1.62 | 82.34±4.92 | 90.23±3.03 | 2.06±4.71 | 14.18±44.01 | 90.12±3.06 |
| uC(stage2) | 30.28M | 330.01G | 89.73±3.12 | 94.37±1.92 | 0.35±0.38 | 2.36±4.69 | 93.94±1.99 | 82.47±5.66 | 90.29±3.48 | 0.69±1.34 | 5.57±18.48 | 90.18±3.50 |
| +DFi | 23.49M | 267.34G | 90.03±3.09 | 94.53±1.93 | 0.49±0.50 | 2.50±5.19 | 94.11±1.99 | 82.76±5.71 | 90.45±3.54 | 1.30±1.83 | 6.05±20.53 | 90.34±3.56 |
| uC(stage3) | 29.92M | 286.09G | 88.71±3.31 | 93.75±2.12 | 0.43±0.42 | 2.70±5.17 | 93.26±2.18 | 81.85±5.77 | 89.91±3.63 | 0.86±1.61 | 5.84±20.54 | 89.79±3.65 |
| +DFi | 23.12M | 223.39G | 89.33±3.30 | 94.13±2.07 | 0.55±0.64 | 2.67±5.30 | 93.68±2.13 | 82.19±6.36 | 90.09±4.02 | 1.41±2.47 | 6.27±21.00 | 89.97±4.04 |
| uC(stage1,2,3) | 45.33M | 527.75G | 90.52±2.77 | 94.84±1.68 | 0.35±0.33 | 2.20±4.07 | 94.43±1.76 | 82.66±5.63 | 90.40±3.48 | 0.76±1.16 | 5.11±16.15 | 90.29±3.51 |
| +DFi | 38.53M | 465.07G | 90.86±2.75 | 95.03±1.68 | 0.34±0.45 | 2.34±5.23 | 94.63±1.73 | 83.68±5.31 | 91.03±3.24 | 0.70±1.66 | 5.73±20.69 | 90.92±3.26 |

| Retinal mIOU | Dice | ASSD | HD95 | AdjRand | Macular Edema mIOU | Dice | ASSD | HD95 | AdjRand | Choroid mIOU | Dice | ASSD | HD95 | AdjRand |
|---|---|---|---|---|---|---|---|---|---|---|---|---|---|---|
| 98.01±1.52 | 98.99±0.79 | 0.14±0.13 | 1.12±0.43 | 98.33±1.31 | 84.74±8.36 | 91.51±5.34 | 0.23±0.21 | 1.16±0.74 | 91.40±5.35 | 94.99±1.84 | 97.42±0.98 | 2.76±7.33 | 23.52±76.65 | 96.64±1.25 |
| 97.97±1.45 | 98.97±0.76 | 0.15±0.12 | 1.12±0.42 | 98.30±1.25 | 86.04±6.84 | 92.34±4.19 | 0.21±0.22 | 1.02±0.72 | 92.24±4.20 | 95.32±1.25 | 97.60±0.66 | 0.30±0.15 | 1.48±0.78 | 96.85±0.96 |
| 97.93±1.60 | 98.95±0.84 | 0.15±0.14 | 1.15±0.51 | 98.26±1.38 | 83.70±8.09 | 90.91±5.13 | 0.22±0.22 | 1.19±0.67 | 90.79±5.14 | 94.82±1.49 | 97.33±0.79 | 0.32±0.15 | 1.52±0.77 | 96.52±1.12 |
| 97.90±1.51 | 98.93±0.79 | 0.15±0.13 | 1.15±0.50 | 98.24±1.30 | 84.21±8.35 | 91.18±5.44 | 0.23±0.22 | 1.25±1.11 | 91.07±5.44 | 95.24±1.40 | 97.56±0.74 | 0.30±0.16 | 1.55±0.75 | 96.80±1.09 |
| 97.70±1.59 | 98.83±0.83 | 0.17±0.14 | 1.15±0.53 | 98.07±1.37 | 81.27±9.28 | 89.36±6.21 | 0.26±0.21 | 1.37±0.94 | 89.22±6.22 | 94.04±2.07 | 96.92±1.11 | 0.45±0.32 | 2.43±2.12 | 95.97±1.53 |
| 97.85±1.53 | 98.91±0.80 | 0.15±0.13 | 1.15±0.50 | 98.20±1.32 | 82.76±7.92 | 90.34±5.10 | 0.28±0.25 | 1.21±0.61 | 90.22±5.10 | 94.53±1.70 | 97.18±0.91 | 0.38±0.23 | 2.05±1.56 | 96.33±1.18 |
| 98.03±1.48 | 99.00±0.77 | 0.13±0.12 | 1.12±0.43 | 98.35±1.27 | 85.94±6.96 | 92.29±4.28 | 0.22±0.22 | 1.02±0.51 | 92.18±4.28 | 95.44±1.60 | 97.66±0.84 | 0.29±0.19 | 1.55±1.15 | 96.92±1.30 |
| 98.03±1.49 | 99.00±0.78 | 0.14±0.12 | 1.12±0.43 | 98.35±1.29 | 86.46±7.87 | 92.54±4.94 | 0.22±0.23 | 1.00±0.78 | 92.44±4.95 | 95.27±1.62 | 97.57±0.85 | 0.30±0.14 | 1.51±0.70 | 96.82±1.16 |

Table 21: The detailed results of the analysis experiments evaluating parameters and performance of 3D UX-Net across various channel depths and the integration of uC on the FLARE2021 dataset. The best values for each metric are highlighted in bold.

| Method | #Params | FLOPs | IoU | Dice | Average ASSD | HD95 | AdjRand | Liver IoU | Dice | ASSD | HD95 | AdjRand |
|---|---|---|---|---|---|---|---|---|---|---|---|---|
| 3D UX-Net (16) | 6.1M | 72.2G | 87.66±3.45 | 92.77±2.52 | 1.80±2.16 | 9.76±17.32 | 92.66±2.53 | 96.06±1.49 | 97.98±0.78 | 1.83±4.00 | 10.81±39.39 | 97.75±0.90 |
| 3D UX-Net (16) +uC | 6.7M | 97.1G | 88.09±2.36 | 93.12±1.59 | 1.05±0.61 | 2.99±1.28 | 93.01±1.61 | 96.41±1.16 | 98.17±0.60 | 1.55±0.49 | 1.55±0.49 | 97.96±0.70 |
| 3D UX-Net (24) | 13.4M | 160.2G | 88.19±2.54 | 93.24±1.64 | 1.63±2.11 | 7.46±11.84 | 93.12±1.67 | 95.40±3.75 | 97.61±2.09 | 2.57±7.19 | 13.50±43.47 | 97.35±2.25 |
| 3D UX-Net (24) +uC | 15.4M | 220.2G | 88.75±2.69 | 93.53±1.97 | 0.92±0.59 | 2.74±1.44 | 93.43±1.98 | 96.48±1.36 | 98.21±0.71 | 0.53±0.30 | 1.57±0.58 | 98.00±0.83 |
| 3D UX-Net (48) | 53.0M | 639.4G | 88.62±2.90 | 93.43±1.98 | 2.59±6.00 | 13.27±46.73 | 93.32±2.00 | 96.48±1.33 | 98.20±0.69 | 1.49±2.83 | 9.76±35.47 | 98.00±0.79 |
| 3D UX-Net (48) +uC | 60.6M | 867.91G | 89.62±2.31 | 94.11±1.52 | 0.76±0.35 | 2.53±1.34 | 94.01±1.55 | 96.73±1.24 | 98.34±0.64 | 0.46±0.25 | 1.47±0.46 | 98.14±0.76 |

| Kidney IoU | Dice | ASSD | HD95 | AdjRand | Spleen IoU | Dice | ASSD | HD95 | AdjRand | Pancreas IoU | Dice | ASSD | HD95 | AdjRand |
|---|---|---|---|---|---|---|---|---|---|---|---|---|---|---|
| 93.12±2.87 | 96.41±1.55 | 1.47±3.41 | 12.89±45.20 | 92.87±3.21 | 95.33±2.02 | 97.60±1.07 | 1.76±3.20 | 8.98±23.76 | 92.56±3.18 | 66.15±11.09 | 79.07±8.72 | 2.13±1.23 | 6.35±4.13 | 78.98±8.74 |
| 92.87±3.21 | 96.28±1.73 | 0.66±0.45 | 2.63±2.54 | 96.19±1.78 | 95.91±1.84 | 97.90±0.97 | 0.39±0.27 | 1.29±0.76 | 97.87±0.98 | 67.17±7.47 | 80.13±5.44 | 2.64±2.04 | 6.50±3.90 | 80.03±5.45 |
| 93.48±3.10 | 96.60±1.67 | 0.68±0.61 | 2.57±2.98 | 96.52±1.72 | 95.37±2.73 | 97.61±1.47 | 1.14±1.90 | 7.62±21.20 | 92.74±2.21 | 68.52±6.56 | 81.15±4.71 | 2.13±1.28 | 6.15±3.42 | 81.05±4.73 |
| 93.04±3.11 | 96.37±1.68 | 0.67±0.49 | 2.56±2.65 | 96.28±1.72 | 96.05±1.48 | 97.98±0.77 | 0.81±1.44 | 1.16±0.28 | 97.95±0.78 | 69.43±9.29 | 81.59±7.21 | 1.68±0.99 | 5.68±4.40 | 81.50±7.23 |
| 93.87±3.01 | 96.81±1.62 | 0.62±0.46 | 2.27±2.56 | 96.74±1.66 | 95.73±1.84 | 97.81±0.97 | 3.28±9.21 | 20.42±83.86 | 97.77±0.98 | 68.40±9.31 | 80.88±6.78 | 4.98±12.51 | 20.64±67.90 | 80.79±6.81 |
| 94.05±2.90 | 96.91±1.56 | 0.63±0.46 | 2.29±2.52 | 96.84±1.60 | 96.34±1.33 | 98.13±0.69 | 0.51±0.48 | 1.13±0.23 | 98.10±0.70 | 71.35±7.46 | 83.06±5.27 | 1.45±0.63 | 5.25±4.41 | 82.98±5.29 |

Table 22: The detailed results of the analysis experiments evaluating parameters and performance of 3D UX-Net across various channel depths and the integration of uC on the OIMHS dataset. The best values for each metric are highlighted in bold.

| Method | #Params | FLOPs | IoU | Dice | Average ASSD | HD95 | AdjRand | Macular Hole IoU | Dice | ASSD | HD95 | AdjRand |
|---|---|---|---|---|---|---|---|---|---|---|---|---|
| 3D UX-Net (16) | 6.1M | 72.2G | 86.06±3.36 | 92.15±2.19 | 1.12±0.90 | 5.79±10.80 | 91.54±2.26 | 77.11±6.52 | 86.93±4.29 | 2.89±3.32 | 16.77±43.16 | 86.78±4.30 |
| 3D UX-Net (16) +uC | 6.7M | 97.1G | 89.60±3.19 | 94.28±2.00 | 0.36±0.40 | 2.47±4.95 | 93.84±2.07 | 81.48±6.53 | 89.66±4.08 | 0.76±1.55 | 5.97±19.84 | 89.54±4.10 |
| 3D UX-Net (24) | 13.4M | 160.2G | 87.86±3.15 | 93.27±1.95 | 0.64±0.67 | 3.68±6.63 | 92.74±2.03 | 79.58±5.88 | 88.51±3.71 | 1.23±1.89 | 6.70±22.04 | 88.38±3.73 |
| 3D UX-Net (24) +uC | 15.4M | 220.2G | 90.23±2.82 | 94.67±1.72 | 0.39±0.44 | 2.37±5.09 | 94.25±1.79 | 82.48±5.32 | 90.31±3.22 | 0.89±1.65 | 5.80±20.19 | 90.19±3.25 |
| 3D UX-Net (48) | 53.0M | 639.4G | 88.35±3.23 | 93.56±2.04 | 0.51±0.48 | 2.70±4.86 | 93.05±2.09 | 80.12±6.29 | 88.83±4.03 | 1.07±1.74 | 6.00±19.30 | 88.71±4.03 |
| 3D UX-Net (48) +uC | 60.6M | 867.91G | 90.38±3.47 | 94.73±2.15 | 0.34±0.42 | 2.39±5.00 | 94.33±2.22 | 82.20±6.23 | 90.10±3.86 | 0.74±1.53 | 5.93±19.73 | 89.99±3.89 |

| Retinal IoU | Dice | ASSD | HD95 | AdjRand | Macular Edema IoU | Dice | ASSD | HD95 | AdjRand | Choroid IoU | Dice | ASSD | HD95 | AdjRand |
|---|---|---|---|---|---|---|---|---|---|---|---|---|---|---|
| 97.35±1.62 | 98.65±0.85 | 0.27±0.29 | 1.15±0.53 | 97.77±1.41 | 77.92±8.83 | 87.30±5.93 | 0.70±1.12 | 2.01±3.53 | 87.13±5.93 | 91.85±2.56 | 95.73±1.40 | 0.64±0.44 | 3.23±2.84 | 94.46±1.92 |
| 97.87±1.45 | 98.92±0.75 | 0.16±0.13 | 1.12±0.43 | 98.21±1.25 | 84.03±8.11 | 91.10±5.22 | 0.20±0.15 | 1.08±0.57 | 90.99±5.23 | 95.00±1.60 | 97.43±0.85 | 0.33±0.17 | 1.69±0.92 | 96.62±1.21 |
| 97.73±1.59 | 98.85±0.83 | 0.20±0.16 | 1.15±0.53 | 98.10±1.38 | 81.05±7.90 | 89.32±5.02 | 0.59±1.60 | 4.37±15.58 | 89.18±5.03 | 93.06±2.22 | 96.39±1.20 | 0.54±0.32 | 2.50±1.84 | 95.29±1.66 |
| 97.96±1.46 | 98.96±0.76 | 0.15±0.12 | 1.12±0.43 | 98.29±1.26 | 85.33±7.84 | 91.88±4.95 | 0.21±0.20 | 1.12±0.83 | 91.77±4.96 | 95.15±1.51 | 97.51±0.80 | 0.29±0.14 | 1.45±0.70 | 96.75±1.08 |
| 97.81±1.54 | 98.89±0.80 | 0.16±0.13 | 1.15±0.53 | 98.16±1.33 | 82.08±7.93 | 89.94±5.06 | 0.34±0.46 | 1.21±0.75 | 89.81±5.06 | 93.40±2.06 | 96.57±1.11 | 0.47±0.28 | 2.43±1.98 | 95.53±1.53 |
| 98.08±1.48 | 99.03±0.77 | 0.14±0.13 | 1.08±0.49 | 98.39±1.28 | 85.97±8.26 | 92.23±5.20 | 0.20±0.21 | 1.07±0.90 | 92.13±5.21 | 95.27±1.56 | 97.57±0.82 | 0.29±0.15 | 1.47±0.81 | 96.83±1.11 |

Table 23: Detailed results of the ablation study on different stage selections for uC integration in 3D U-Net. The best values for each metric are highlighted in bold.

| Method | IoU | Dice | Average ASSD | HD95 | AdjRand | Macular Hole IoU | Dice | ASSD | HD95 | AdjRand |
|---|---|---|---|---|---|---|---|---|---|---|
| 3D U-Net | 87.08±3.32 | 92.79±2.09 | 1.31±1.35 | 7.39±14.69 | 92.22±2.15 | 77.89±6.74 | 87.42±4.28 | 3.94±5.25 | 24.03±58.65 | 87.28±4.30 |
| +uC (stage 1) | 90.06±3.41 | 94.55±2.16 | 0.96±1.85 | 7.93±19.51 | 94.14±2.20 | 82.49±6.14 | 90.28±3.85 | 0.72±1.68 | 5.89±20.82 | 90.17±3.88 |
| +uC (stage 2) | 89.73±3.12 | 94.37±1.92 | 0.35±0.38 | 2.36±4.69 | 93.94±1.99 | 82.47±5.66 | 90.29±3.48 | **0.69±1.34** | 5.57±18.48 | 90.18±3.50 |
| +uC (stage 3) | 88.71±3.31 | 93.75±2.12 | 0.43±0.42 | 2.70±5.17 | 93.26±2.18 | 81.85±5.77 | 89.91±3.63 | 0.86±1.61 | 5.84±20.54 | 89.79±3.65 |
| +uC (stage 5) | 87.32±3.46 | 92.93±2.27 | 0.69±0.63 | 3.05±5.19 | 92.37±2.31 | 78.68±6.11 | 87.93±4.01 | 1.44±1.92 | 6.65±20.58 | 87.80±4.01 |
| +uC (stage 1, 2, 3) | 90.52±2.77 | 94.84±1.68 | 0.35±0.33 | **2.20±4.07** | 94.43±1.76 | 82.66±5.63 | 90.40±3.48 | 0.76±1.16 | **5.11±16.15** | 90.29±3.51 |
| +uC (stage 1, 2, 3) + DFI | **90.86±2.75** | **95.03±1.68** | **0.34±0.45** | 2.34±5.23 | **94.63±1.73** | **83.68±5.31** | **91.03±3.24** | 0.70±1.66 | 5.73±20.69 | **90.92±3.26** |

| Retinal IoU | Dice | ASSD | HD95 | AdjRand | Macular Edema IoU | Dice | ASSD | HD95 | AdjRand | Choroid IoU | Dice | ASSD | HD95 | AdjRand |
|---|---|---|---|---|---|---|---|---|---|---|---|---|---|---|
| 97.68±1.52 | 98.82±0.80 | 0.17±0.13 | 1.15±0.51 | 98.05±1.32 | 80.39±8.28 | 88.89±5.38 | 0.54±1.06 | 1.52±1.44 | 88.75±5.38 | 92.36±2.43 | 96.01±1.32 | 0.58±0.29 | 2.85±1.97 | 94.83±1.72 |
| 98.01±1.52 | 98.99±0.79 | 0.14±0.13 | **1.12±0.43** | 98.33±1.31 | 84.74±8.36 | 91.51±5.34 | 0.23±0.21 | 1.16±0.74 | 91.40±5.35 | 94.99±1.84 | 97.42±0.98 | 2.76±7.33 | 23.52±76.65 | 96.64±1.25 |
| 97.93±1.60 | 98.95±0.84 | 0.15±0.14 | 1.15±0.51 | 98.26±1.38 | 83.70±8.09 | 90.91±5.13 | **0.22±0.22** | 1.19±0.67 | 90.79±5.14 | 94.82±1.49 | 97.33±0.79 | 0.32±0.15 | 1.52±0.77 | 96.52±1.12 |
| 97.70±1.59 | 98.83±0.83 | 0.17±0.14 | 1.15±0.53 | 98.07±1.37 | 81.27±9.28 | 89.36±6.21 | 0.26±0.21 | 1.37±0.94 | 89.22±6.22 | 94.04±2.07 | 96.92±1.11 | 0.45±0.32 | 2.43±2.12 | 95.97±1.53 |
| 97.70±1.53 | 98.83±0.80 | 0.17±0.13 | 1.15±0.51 | 98.07±1.32 | 80.42±8.99 | 88.86±6.10 | 0.49±1.15 | 1.47±1.14 | 88.71±6.09 | 92.48±2.27 | 96.08±1.23 | 0.64±0.55 | 2.94±2.36 | 94.88±1.70 |
| **98.03±1.48** | **99.00±0.77** | **0.13±0.12** | **1.12±0.43** | **98.35±1.27** | 85.94±6.96 | 92.29±4.28 | **0.22±0.22** | 1.02±0.51 | 92.18±4.28 | **95.44±1.60** | **97.66±0.84** | **0.29±0.19** | 1.55±1.15 | **96.92±1.30** |
| 98.03±1.49 | 99.00±0.78 | 0.14±0.12 | 1.12±0.43 | 98.35±1.29 | 86.46±7.87 | 92.54±4.94 | 0.22±0.23 | 1.00±0.78 | 92.44±4.95 | 95.27±1.62 | 97.57±0.85 | 0.30±0.14 | 1.51±0.70 | 96.82±1.16 |

Table 24: Detailed results of the ablation study on different channel depth configurations of uC 3DU-Net on the OIMHS dataset. The best values for each metric are highlighted in bold.

| Method | #Params | FLOPs | IoU | Dice | Average ASSD | HD95 | AdjRand | Macular Hole IoU | Dice | ASSD | HD95 | AdjRand |
|---|---|---|---|---|---|---|---|---|---|---|---|---|
| 3D U-Net | 4.8M | 135.9G | 87.08±3.32 | 92.79±2.09 | 1.31±1.35 | 7.39±14.69 | 92.22±2.15 | 77.89±6.74 | 87.42±4.28 | 3.94±5.25 | 24.03±58.65 | 87.28±4.30 |
| uC 3DU-Net (16) | 5.3M | 134.3G | 89.98±2.69 | 94.53±1.61 | 0.63±0.63 | 2.44±5.06 | 94.11±1.68 | 82.31±5.58 | 90.20±3.38 | 1.75±2.47 | 6.10±20.11 | 90.08±3.41 |
| uC 3DU-Net (24) | 21.7M | 286.4G | 90.67±2.95 | 94.91±1.84 | 0.34±0.46 | 2.35±5.22 | 94.53±1.89 | 83.57±5.21 | 90.96±3.13 | 0.80±1.69 | 5.80±20.64 | 90.86±3.17 |
| uC 3DU-Net (32) | 38.5M | 465.1G | **90.86±2.75** | **95.03±1.68** | **0.34±0.45** | **2.34±5.23** | **94.63±1.73** | **83.68±5.31** | **91.03±3.24** | **0.70±1.66** | **5.73±20.69** | **90.92±3.26** |

| Retinal IoU | Dice | ASSD | HD95 | AdjRand | Macular Edema IoU | Dice | ASSD | HD95 | AdjRand | Choroid IoU | Dice | ASSD | HD95 | AdjRand |
|---|---|---|---|---|---|---|---|---|---|---|---|---|---|---|
| 97.68±1.52 | 98.82±0.80 | 0.17±0.13 | 1.15±0.51 | 98.05±1.32 | 80.39±8.28 | 88.89±5.38 | 0.54±1.06 | 1.52±1.44 | 88.75±5.38 | 92.36±2.43 | 96.01±1.32 | 0.58±0.29 | 2.85±1.97 | 94.83±1.72 |
| 97.95±1.56 | 98.96±0.81 | 0.14±0.13 | 1.15±0.51 | 98.28±1.34 | 84.86±6.96 | 91.66±4.23 | 0.21±0.17 | 1.04±0.50 | 91.54±4.25 | 94.80±1.90 | 97.32±1.02 | 0.44±0.34 | 1.47±0.57 | 96.52±1.28 |
| 97.99±1.51 | 98.98±0.79 | 0.14±0.13 | 1.12±0.43 | 98.32±1.30 | 85.16±8.18 | 91.76±5.26 | 0.21±0.22 | 1.25±1.13 | 91.65±5.26 | **95.95±1.18** | **97.93±0.62** | **0.23±0.10** | **1.22±0.41** | **97.30±0.83** |
| **98.03±1.49** | **99.00±0.78** | **0.14±0.12** | **1.12±0.43** | **98.35±1.29** | 86.46±7.87 | 92.54±4.94 | 0.22±0.23 | 1.00±0.78 | 92.44±4.95 | 95.27±1.62 | 97.57±0.85 | 0.30±0.14 | 1.51±0.70 | 96.82±1.16 |

# D    Qualitative Results

In this section, we present additional qualitative results. Fig. 6, Fig. 7, Fig. 8, and Fig. 9 further show the superior segmentation accuracy of the proposed uC 3D U-Net compared to previous models on the FLARE2021, FeTA2021, OIMHS, and AbdomenCT-1K datasets. Fig. 10 visualizes the impact of integrating the uC module with backbone models on the FLARE2021 dataset. These visualizations demonstrate that the integration of uC effectively enhances the model's ability to extract axial-slice plane information while retaining effective volumetric feature extraction, addressing challenges related to anisotropic medical imaging. This integration achieves improved performance with reduced computational cost and fewer parameters, resulting in an optimal parameter-to-performance ratio compared to previous models.

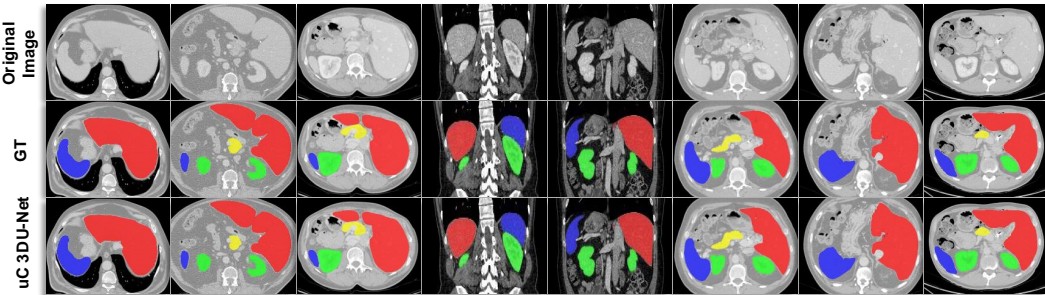

Figure 6: More visual results of the proposed uC 3DU-Net on the FLARE2021 dataset. For enhanced visual clarity, the displayed images have been cropped. Please kindly zoom in for a better view.

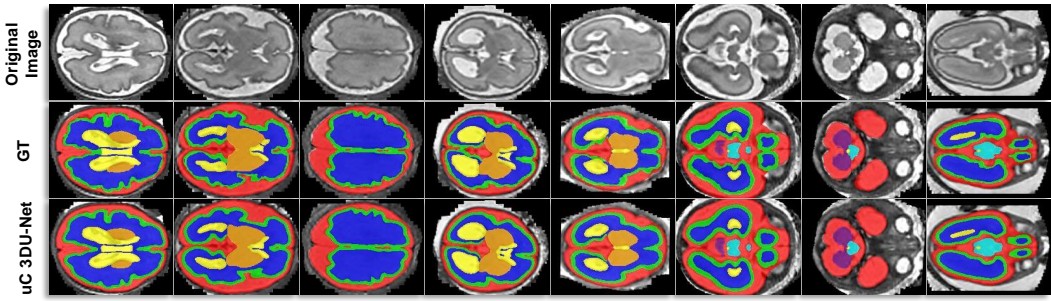

Figure 7: More visual results of the proposed uC 3DU-Net on the FeTA2021 dataset. For enhanced visual clarity, the displayed images have been cropped. Please kindly zoom in for a better view.

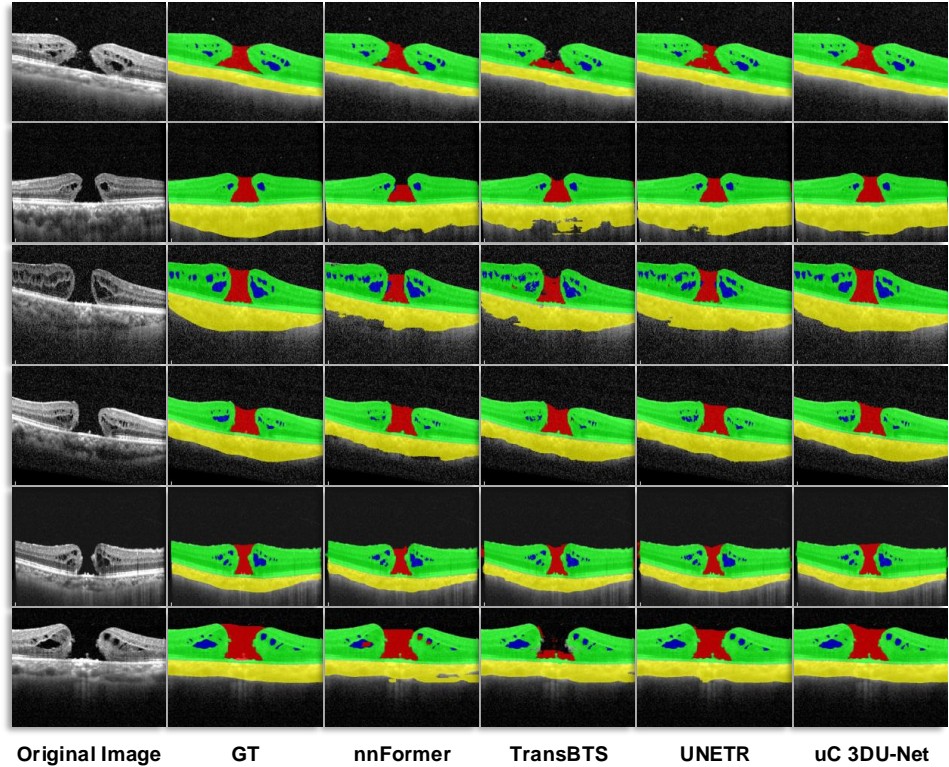

Figure 8: The visual comparison of the validation results on the OIMHS dataset for uC 3DU-Net, and 3 previous segmentation methods. We have selected 6 representative sequences for display. For enhanced visual clarity, the displayed images have been cropped. Please zoom in for a better view.

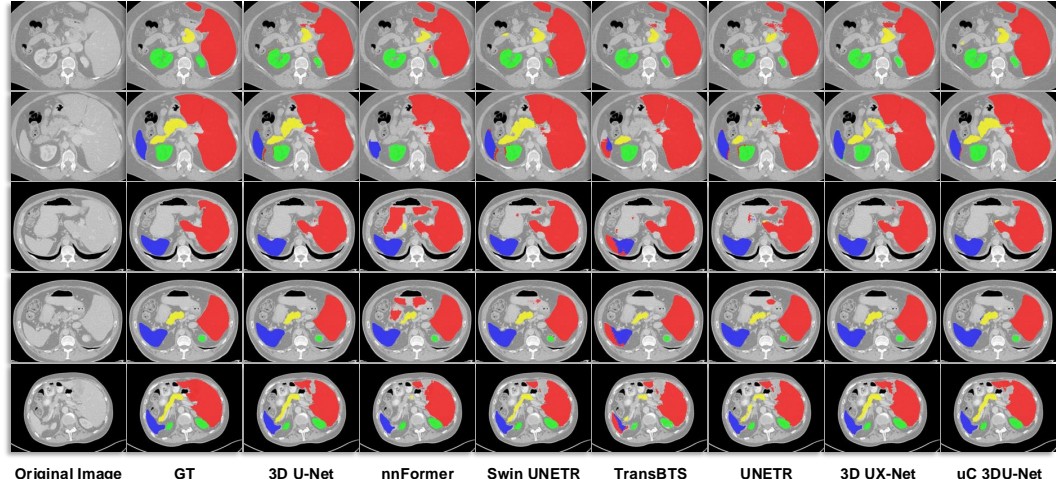

Figure 9: The visual comparison of the validation results on the ABCT1K dataset for uC 3DU-Net, and 6 previous segmentation methods. We have selected 5 representative sequences for display. For enhanced visual clarity, the displayed images have been cropped. Please zoom in for a better view.

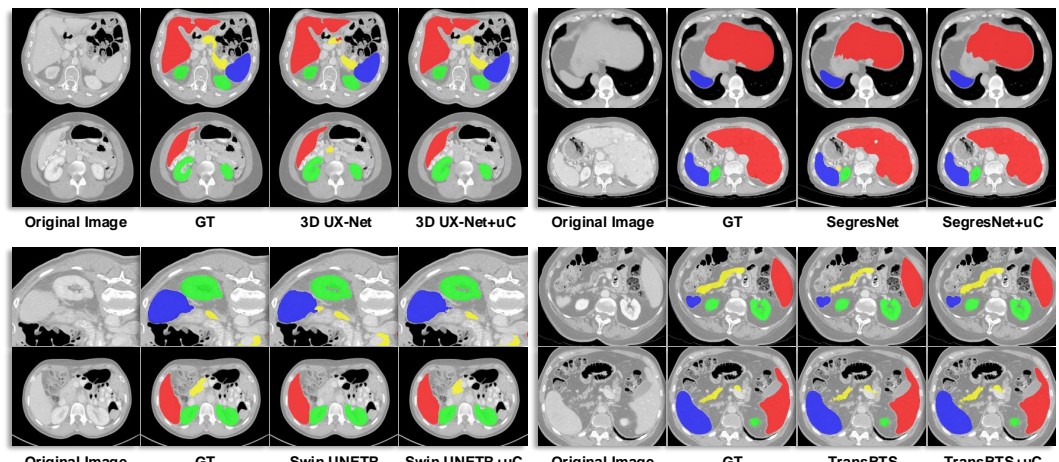

Figure 10: Visual results showcasing the differences in integrating uC for 3D UX-Net, SegResNet, SwinUNETR, and TransBTS. For each model, We have selected two representative sequences for display. For enhanced visual clarity, the displayed images have been cropped. Please zoom in for a better view.

