# OpenReview forum: "Upping the Game: How 2D U-Net Skip Connections Flip 3D Segmentation"
_NeurIPS.cc/2024/Conference — NeurIPS 2024 poster_

### Official Review · Reviewer_7CRU · 2024-07-09

**Soundness:** 3
**Presentation:** 3
**Contribution:** 3
**Rating:** 7
**Confidence:** 3

**Summary:**

This paper proposes a novel element for 3D medical image segmentation, called uC (U-shaped connection) that replaces standard skip connection with 2D convolutions.

**Strengths:**

- uC 3DU-Net shows state-of-the-art results on 4 datasets.
- It reduces computational complexity and number of parameters
- It is aimed at addressing the axial-slice performance drop-off in 3D convolutions

**Weaknesses:**

- Paper is hard to read at times. The language is sometimes complex, using rare words I have never seen used in that context.
- Section 3.2 is essentially empty: there is no clear description of the U-Shaped connection.

**Questions:**

The authors claim their code is available on github. However, I did not find the code, nor a link at https://anonymous.4open.science or some similar one.

Line 246, the reference to AdamW is missing.
Line 182, why 5D volumetric data image?

**Limitations:**

The authors have not highlighted any limitations, nor potential negative societal impact.

---

> ### Author Rebuttal · Authors · 2024-08-06
>
> Thank you for the thoughtful review of our work! Please allow us to address your concerns and answer the questions.
>
> >W1: Paper is hard to read at times.
>
> Your suggestions are pivotal, we will comprehensively optimize the readability of the manuscript.
> >W2: Section 3.2 is essentially empty...
> Section 3.2 has been reorganized and rewritten to reduce unnecessary background information and introduce more detailed descriptions of U-shaped Connection(uC), as follows:
>
> Considering the objective of skip connections to capture detailed features of original images, it is essential to note that 3D medical images fundamentally represent a sequence of 2D images. Given the anisotropic nature of medical images, the 2D slice plane shows richer feature information compared to the temporal sequence axis. Although basic skip connection methods like cat and addition can supplement original feature information to some extent, these methods fail to fully utilize the rich slice plane information in 3D medical images. To address the challenges posed by the anisotropy of 3D medical images, the U-shaped Connection(uC) is proposed. This approach employs a simplified 2D U-Net to implement skip connections, thereby supplementing the rich original 2D axial plane feature information, and enhancing the 2D axial plane feature extraction capabilities of any 3D image segmentation network. The uC combines features extracted by 2D convolution with those extracted by 3D upsampling, offering more efficient feature extraction compared to pure 3D convolution, as detailed structure in Sec3.3.
>
> The fundamental structure of uC is based on a 2D U-Net.  In comparison to the basic 2D U-Net, uC omits the initial and final conv layers, retaining only the downsampling and upsampling layers to minimize computational load and enhance the extraction efficiency of original slice plane features.  Each downsampling layer comprises an average pooling layer and two conv layers, each consisting of 2D conv, Group Normalization (GN), and ReLU.  The average pooling layer reduces the feature map size by half, and the subsequent conv layers double the channel number, with the group number of GN set to half the current channel number.  Each upsampling layer includes a transposed convolution layer and two conv layers, where the transposed convolution layer restores the feature map to its original size, and the conv layers reduce the channel number back to its half.
>
> >Q1: The code.
>
> We have provided the project code on anonymous github and sent the anonymous link to AC.
> >Q2: The reference to AdamW is missing.
>
> Your meticulous review is greatly appreciated. The AdamW citation on line 246 will be rectified in the future version.
>
> >Q3: why 5D volumetric data image?
>
> Thank you for the valuable feedback regarding the description of 5D volumetric data image, which should indeed be 3D volumetric data image on line 182, we will rectified it in the future version.
>
> Your comments have greatly encouraged us, and we are sincerely grateful for the significant help you have provided in refining our manuscript.~ （ ´▽` )ノﾞ

---

> > ### Comment · Reviewer_7CRU · 2024-08-12
> >
> > Thank you for your response.
> >
> > I believe you could have added the anonymous GitHub link to the response, for everyone and not just for the AC.
> >
> > At this stage, I see no reason to change my score.

---

### Official Review · Reviewer_J8Kk · 2024-07-09

**Soundness:** 3
**Presentation:** 4
**Contribution:** 3
**Rating:** 7
**Confidence:** 4

**Summary:**

The manuscript proposes using 2D U-Net connections to enhance 3D U-Net segmentation. The innovative approach demonstrates excellent performance across different imaging modalities and achieves state-of-the-art (SOTA) results with fewer parameters and lower computational costs on four public datasets. The manuscript thoroughly describes the challenges associated with 3D segmentation.

**Strengths:**

1. The creative use of 2D U-shape connections in a 3D network shows impressive performance.
2. The method achieves SOTA performance with fewer parameters and lower computational costs across four public datasets.
3. The manuscript provides a comprehensive description of the challenges encountered in 3D segmentation.

**Weaknesses:**

1. The principle behind the proposed DFI is not well-explained.
2. In the ablation studies section, the HD95 metric for +cU(stage 1, 2, 3)+DFI is slightly higher than for +cU(stage 1, 2, 3).

**Questions:**

1. Can authors discuss more about the principle of DFI?
2. Why is the HD95 metric for +cU(stage 1, 2, 3)+DFI slightly higher than for +cU(stage 1, 2, 3)?

**Limitations:**

The impact of DFI on various 3D segmentation backbones is not explored.

---

> ### Author Rebuttal · Authors · 2024-08-06
>
> Thank you for your insightful comments to our work! We appreciate the opportunity to address your concerns and respond to your questions.
>
> **W1 & Q1:**
> The motivation for employing DFi and its underlying principles can be futher discribed: In conventional 3D U-Net architectures, skip connections integrate the original feature information from the encoder during the upsampling process through cat, followed by two 3x3 convolutions to reduce the number of channels and consolidate useful information. However, this rudimentary feature fusion approach may lead to inefficiencies when reconciling the disparity between 3D spatial features and 2D slice plane features. Therefore, we devise a streamlined, efficient multi-scale feature fusion module DFi. The 2D slice plane features extracted by uC and the 3D spatial features from the 3D U-Net encoder are cat to retain maximal original feature information.                     Howerver, subsequent convolution operations to reduce channel numbers are computationally intensive and inefficient. Utilizing addition for feature fusion could result in information loss due to inherent discrepancies between the two types of features.                     Hence, an approach combining cat, addition, and subsequent convolution layers, with the aim of minimizing parameter count and FLOPs, resulting in the design of the DFi module.
> For the 2D slice plane features F1 extracted by uC and the 3D spatial features F2 obtained during 3D U-Net upsampling, branch 1 of the DFi module cats F1 and F2, then employs a 1x1 conv to halve the channel count, preserving salient features to produce feature map Fc. Branch 2 initially extracts critical information from F1 and F2 using 1x1 convs, followed by addition and sigmoid activation to generate the attention map Fa. The element-wise multiplication of Fc and Fa adjusts and integrates the feature map Fc from branch 1, thereby learning the relative importance of F1 and F2 during the fusion process.
> The proposed DFi module integrates and simplifies addition, cat, and subsequent
> convolution layers, achieving a more lightweight and efficient fusion of 2D and 3D spatial features. Additionally, we conducted supplementary experiments to validate the efficacy of DFi and compared its parameter count and FLOPs, as shown in pdf.Tab.6.                                 The experimental results indicate that employing DFi, as opposed to direct cat, offers a performance improvement with a reduced parameter count and FLOPs, thereby justifying  its replacement of the concatenation operation in the 3D U-Net is helpful.
>
>
> **W2 & Q2:**
> As you noted, while DFi induces a slight increase in HD95, accompanied by modest improvements in mIoU and Dice metrics, the overall performance impact remains positive. This is corroborated by the additional experiments presented in pdf.Tab.6. Given that DFi offers a computationally efficient feature fusion method in the uC 3D U-Net, its substitution for concatenation is advantageous.
>
> We are truly grateful for your insightful comments and the time you have invested in improving the quality of our U-shaped Connection. The feedback is highly motivating, and we appreciate your valuable comments! ヽ( ´▽` )ノ

---

> > ### Comment · Reviewer_J8Kk · 2024-08-12
> >
> > Thank you for your response. However, while your rebuttal has clarified certain aspects of the work, it does not alter my overall assessment of the paper. Therefore, I have decided to maintain my original score.

---

### Official Review · Reviewer_WWao · 2024-07-13

**Soundness:** 2
**Presentation:** 2
**Contribution:** 2
**Rating:** 5
**Confidence:** 5

**Summary:**

The study introduces uC 3DU-Net, a 3D medical image segmentation method combining 2D U-Net skip connections with 3D CNNs.

**Strengths:**

- Experiments were conducted across multiple datasets with several relevant baselines.
- Conducted comprehensive ablation studies and model hyperparameters are reported.

**Weaknesses:**

**Motivation:** The justification provided for the selection of a 2D UNet over 3D convolutions in Section 3.2 is somewhat insufficient. The authors argue that 3D convolutions are less efficient at extracting 2D plane information; however, this assertion does not consider that using 2D convolutions on 3D medical images can compromise the spatial relationships between slices, thereby limiting the capture of intra-slice features. Could the authors elaborate on their rationale behind this choice? Furthermore, the novelty of the proposed approach appears primarily limited to the incorporation of a 2D UNet as a skip connection. A substantial portion of the paper (approximately four pages) is devoted to discussing previous methodologies, while the method is described only in Section 3.3 and this makes it hard to see the contributions of this work among the extensive discussion of other research.

Additionally, the reference to 3D convolutions in Line 195 seems inconsistent with the dedicated use of a 2D UNet and requires further clarification. The main figure (Figure 2) also lacks a detailed representation of the blocks used in the 3D UNet, which obscures understanding of the architecture.

**Experiments:** To robustly support the choice of a 2D UNet, a comparative analysis with at least one 3D UNet or 3D convolutional model would be highly informative. This would provide a stronger justification for the choice of model architecture.  Furthermore, testing the models on more challenging datasets, such as the BTCV 13 class dataset, would better demonstrate their efficacy and robustness. There is also a discrepancy between the number of parameters listed for SegResNet in Table 1 and Table 4 that needs to be addressed for clarity and accuracy in the reported results.

Since the addition of the DFi module results in only minor improvements, the performance gains claimed could be attributed to random variation, given the very small difference in the provided metrics. This minimal change does not convincingly demonstrate the effective integration of 2D and 3D features.

**Minor comments:**
Fix the AdamW citation in line 246.

**Questions:**

Please visit Weaknesses.

**Limitations:**

Please visit Weaknesses.

---

> ### Author Rebuttal · Authors · 2024-08-06
>
> Thank you for your careful review of our work! We would like to take this chance to address your concerns and respond to your questions.
> W-Motivation:
> >W1: The justification provided for...
>
> The utilization of 2D convolutions potentially undermines spatial relationships between slices in 3D imaging; thus, comprehensive experiments are imperative to ascertain both the enhancement of planar performance and the impact on 3D spatial relationships. Extensive experiments unequivocally demonstrate a significant improvement in the performance of 2D convolutions with negligible detriment to spatial integrity. CT scans produce independent images through multi-angle X-ray scanning, whereas MRI relies on magnetic fields and RF pulses to excite hydrogen atoms, imaging only at specific resonant slices. Therefore, the spatial disruption by 2D convolutions in 3D sequences is minimal, whereas processing each slice independently is crucial. This hypothesis is substantiated by extensive experimentation:
>
> * Image Reconstruction Fidelity Comparison: Three networks—3D U-Net, 3D U-Net with 3D uC, and 3D U-Net with 2D uC—were employed to reconstruct input images. Evaluation of the reconstructed planar images, as shown in pdf.Tab.4, revealed that the 3D U-Net with 2D uC achieved the highest PSNR, indicating superior feature extraction and reconstruction capabilities compared to 3D convolutions.
>
> * Convolutional Dimensionality Performance Comparison: The relationship between the dice coefficient and parameter quantity for the three networks under different initial channel counts was assessed (pdf.Fig.2). The performance disparity directly attributed to the 2D and 3D convolution efficiency in the uC structure was evident, excluding the influence of skip connections. As shown in pdf.Tab.3 and pdf.Fig.1, the 2D uC network converged more rapidly on the validation set with fewer parameters, demonstrating the superior efficiency of 2D convolutions in planar feature extraction.
>
> * Dimensional Slicing Analysis: Comparison of slicing along three dimensions in the uC structure (pdf.Tab.2) showed that, although the information density varies across dimensions in 3D medical imaging, enhancing 2D information in any plane slightly improves overall performance with minimal spatial relationship disruption. This justifies the extraction of 2D features within 3D segmentation models under the premise of capturing inter-dimensional relationships via 3D convolutions.
>
> >W2: Furthermore, the novelty...
>
> The research primarily addresses the challenges of anisotropy in 3D medical imaging, where isotropic 3D convolutions fail to adequately capture differential information densities across dimensions. As demonstrated in pdf.Tab.4, our experiments on 3D U-Net, 3D U-Net with 2D uC, and 3D U-Net with 3D uC underscore the efficacy of 2D convolutions in enhancing slice-wise feature extraction. uC offers a solution to anisotropy in 3D medical imaging. Additional details on our methodology will be included in the final version of the paper.
> W3: Additionally, the reference to 3D convolutions...
> In the description on lines 194-195, uC1-uC3 represents a simplified 2D U-Net structure rather than a 3D U-Net structure. We apologize for this oversight. We will provide further details on the uC 3D U-Net architecture and the blocks used in Fig.2 and its caption in the final paper version.
>
> W-Experiments:
> >W4: To robustly support the choice of a 2D UNet...
>
> A comparative evaluation involving 3D U-Net, 3D U-Net with 2D uC and 3D uC is detailed in pdf.Tab.3. Figure 2 further isolates the impact of 2D and 3D convolutions within the uC modules by removing the influence of skip connections, thereby attributing observed performance differences directly to convolutional efficacy, where 2D convolution demonstrates superior slice-wise feature extraction compared to 3D. This finding is reinforced by image reconstruction experiments (pdf.Tab.4), where 2D convolution outperforms 3D in image details. The evaluation of 2D uC applied across arbitrary planes (pdf.Tab.2) consistently enhances 3D U-Net's performance, suggesting a suboptimal exploitation of planar information within the three-dimensional architecture.
> >W5: Challenging datasets.
>
> We validated our proposed method on the BTCV standard dataset (pdf.Tab.5). Using 30 annotated cases for training and validation, the method demonstrated significant efficacy. Given the smaller dataset size, CNN-based models outperformed Transformer-based models, with more details to be included in the revised paper.
> >W6: The number of parameters for SegResNet.
>
> The baseline data in pdf.Tab.1 was sourced from the 3D UX-Net paper, which used a smaller initial channel count (8) for SegResNet, resulting in lower parameters and FLOPs. Conversely, pdf.Table 4 reflects a larger initial channel count (32) for enhanced performance, aligning SegResNet's parameters with 3D U-Net.
> >W7: Since the addition of the DFi...
>
> Future versions will include more details of the DFi module. Unlike U-Net's skip connections, which merge encoder features during upsampling via concatenation and subsequent convolutions, the proposed DFi module integrates addition, concatenation, and convolution layers for efficient 2D and 3D feature fusion with reduced computational load. Experimental results in pdf.Tab.6 indicate that DFi offers a slight performance improvement with lower parameter and FLOPs counts compared to direct concatenation. The paper's focus on DFi will shift towards feature fusion efficiency rather than effect. Continuous improvement efforts will aim to enhance 2D and 3D feature integration without increasing computational burden.
> W-Minor comments:
> >W8: Minor comments: Fix the AdamW citation in line 246.
>
> The AdamW citation in line 246 will be modified in the future version.
>
> Thank you once again for your thoughtful review, which has greatly encouraged us! We hope that our revisions will address all your concerns and clear up your concerns.
> ヽ（＾ω＾）ノ

---

> > ### Comment · Reviewer_WWao · 2024-08-12
> > **Official Comment by Reviewer WWao**
> >
> > Thank you for providing the clarification and the requested experiments. Given these clarifications, I will revise my score. However, the paper requires major revision in terms of clearly describing its contributions, as much of the manuscript is devoted to discussing related works.

---

### Official Review · Reviewer_Lrjd · 2024-07-15

**Soundness:** 2
**Presentation:** 3
**Contribution:** 2
**Rating:** 5
**Confidence:** 4

**Summary:**

This paper proposes an approach to combine 3D and 2D features for medical image segmentation. The key observation is that medical images often have high in-plane resolution (2D) and lower through-plane resolution (across 2D slices). They introduce a U-shaped connection which uses 2D convolutions in place of standard skip connections in the widely used UNet architecture. They demonstrate segmentation results on four public datasets across a variety of imaging modalities and organs. Their method is computationally more efficient as there are fewer parameters needed by pushing some computations to 2D.

**Strengths:**

This paper presents a simple and clever modification to existing UNet-based networks for medical image segmentation. The key idea is to add 2D UNet feature extractors to several layers of a 3D UNet to better learn the high resolution in-plane features. They augment these features with the typical 3D features in a DFI block. The idea is simple and flexible, and can be incorporated into existing UNet-based architectures.

They evaluate their proposed approach on four public datasets and compare with 9 baseline architectures. They also include several ablation studies testing their method. They achieve improved results by1-2 Dice points compared to baseline approaches.

**Weaknesses:**

The method is a fairly simple architectural tweak that improves performance very slightly, on already well-studied segmentation tasks. Consequently, there should be substantially more experimental evaluation to properly conclude the benefit of this approach over other methods, especially attention-based approaches.

The main claim of this work is that using an isotropic 3D convolution will ignore or will not accurately learn in-slice features. Additional experiments should have been done to demonstrate this. For example, how would this model perform on a highly nonistropic dataset, or a high-resolution, isotropic one (such as brain MRI)?

Are the table results shown on the test set or the validation set? How was the validation set used?

For the baseline models, I believe additional details should have been provided:

- how was the hyperparameter tuning done? Was there a proper grid search to determine model depth, regularization parameters, etc...? For example, the very poor performance of nnFormer in Table 10 and of TransBTS in Table 11 are concerning.
- The set of data augmentations is quite limited, and may have hurt the models with a larger number of parameters (e.g. transformer-based). Additional nonlinear augmentations should have been performed.

The ablation studies also reveal negligible improvement when adding the uC to an existing model (Table 5). This is perhaps the most revealing experiment of this work.

All experiments should have associated error bars. The differences are far too small to make any meaningful conclusions.

I am also not convinced that the main claim of the paper is proven:
*While 3D convolutions capture sequential spatial feature information, their efficiency in extracting 2D plane information is inferior to that of 2D convolutions applied to individual slice images.* (lines 169-170)

**Questions:**

- How are the images reshaped to be run in the 2D UNet model? Is the 3D image tiled to create a 2D image?
- How would network performance change for a high resolution, isotropic dataset? For a dataset with low out-of-plane resolution?
- Why does the nnUNet model have substantially more parameters than your model?

**Limitations:**

Minimal limitations are mentioned.

---

> ### Author Rebuttal · Authors · 2024-08-06
>
> Thank you for the thoughtful review of our work! Below are the responses to your concerns.
> >W1: More experimental evaluation.
>
> We conducted experiments involving two attention-based segmentation models, D-LKA-former[1] and UNETR++[2], across four datasets, with results presented in pdf.Tab.1. Updated manuscript will include the five-fold cross-validation results for these newly added baselines.
> >W2&Q2: How would this model perform on a highly nonistropic dataset, or a high-resolution, isotropic one?
>
> Comparative experiments were conducted on the anisotropic BTCV standard dataset (in-plane resolution:0.54*0.54-0.98*0.98mm², slice thickness:2.5-5.0mm). The results, shown in pdf.Tab.5, indicate that our method performs favorably on this anisotropic dataset. Additionally, within our datasets, FeTA represents a typical isotropic brain MRI dataset (spatial resolution:0.43-1mm cross all dimensions), while OIMHS is a highly anisotropic dataset (spacing:10.7-14.0 µm in-plane and 7.0-40.0 µm through-plane). The performance improvement of uC on the OIMHS depicted in Tab.1 and 2, exceeds that on the FeTA, underscoring the efficiency of the proposed method in enhancing in-slice plane information extraction, particularly on anisotropic datasets.
> >W3: ...shown on the test set or the validation set? How was the validation set used?
>
> ​All results are conducted on test sets, with data partitioned in an 8:1:1 ratio for training, validation, and testing set. Validation occurred every 1000 iter during training, saving the model weights with the best validation performance. We will include all details in the future version.
> >W4: More details for the baseline models.
>
> We adhered to validated hyperparameter settings in previous papers for baseline models, avoiding additional grid searches for hyperparameters like model depth and regularization. The training protocols and epochs for models like nnFormer and TransBTS matched those from established papers, with code available for reproducibility. Data preprocessing for comparative experiments was consistent with methods from the 3D UX-Net[3] paper. For transformer-based models with large parameter counts, training involved extensive data augmentation techniques on over 300 cases from specific datasets.
> >W5: Experiment about adding the uC to an existing model.
>
> Experiments of uC in pdf.Tab.5 highlight two key insights. First, in fully optimized 3D segmentation tasks, an Dice improvement of 0.43% to 1% is significant, as shown by the FLARE2021, where a 1% gain necessitates a tenfold model increase, indicating the challenge of enhancing performance with convolutional architectures. As pdf.Tab.2, transitioning from 3D U-Net to 3D UX-Net[3] in OIMHS yields only a 0.85% Dice improvement, yet this still validates the efficacy of 3D UX-Net, as recognized in the field. The observed Dice improvements of 0.4%-1% across models such as 3D UX-Net and Swin UNETR[4] confirm this advancement as substantial in 3D segmentation.
>
> Furthermore, the analysis in pdf.Tab.5 demonstrates the method’s cost-effectiveness compared to simply increasing channels. The 3D UX-Net, with 24 initial channels and the uC module, outperforms the 48-channel version while reducing parameters and FLOPs to under 30%, emphasizing the method’s contribution to lightweighting. This efficiency, with 15.4M parameters surpassing a model with 340% more, underscores the method's effectiveness.
> >W6: All experiments should have associated error bars.
>
> The updated manuscript will include standard deviations for all experimental results wherever possible, similar to those provided in pdf.Tab.1.
> >W7: Main claim of the paper...
>
> To address the concern regarding the efficiency of 2D convs in extracting slice-plane information, we have conducted additional experiments:
> * Convolutional Dimensionality Performance Comparison: Visualization results in pdf.Fig.2 demonstrate the relationship between dice scores and parameter counts for three models: 3D U-Net, +3D uC, +2D uC. The performance differences between the 2D and 3D convs in the uC, excluding the impact of skip connections, indicate the superior efficiency of 2D convs in slice-plane feature extraction.
> * Image Reconstruction Fidelity Comparison: We evaluated three networks—3D U-Net, +3D uC, +2D uC—on their ability to reconstruct input images, with the results presented in pdf.Tab.4. The highest PSNR achieved by 3D U-Net+2D uC confirms the proposed 2D uC structure's advantage in extracting slice-plane features.
> * Dimensional Slicing Analysis: Experiments slicing input images along different dimensions reveal that slicing along the plane dimension yields the best results of uC, as shown in pdf.Tab.2. This reflects the denser information present in the Depth dimension of 3D medical images, enhancing performance with the proposed uC approach. These additional experiments collectively verify the enhanced efficiency of 2D convs in extracting in-slice features compared to 3D convs.
>
> **Q1:** Images are reshaped by stacking slices along the batch dimension, implemented in code with 'einops.rearrange(x, 'B C D H W -> (B W) C D H')'.
>
> **Q3:** Parameter comparisons including nnUNet, follow the data reported in the 3D UX-Net[3]. The smaller parameter count for our model attributed to fewer channels and the use of DFi. The pdf.Tab.6 provides detailed parameters comparisons when using DFi.
>
> [1] Azad, Reza, et al. "Beyond self-attention: Deformable large kernel attention for medical image segmentation." WACV. 2024.
>
> [2] Shaker, Abdelrahman M., et al. "UNETR++: delving into efficient and accurate 3D medical image segmentation." IEEE TMI, 2024.
>
> [3] Lee, Ho Hin, et al. "3d ux-net: A large kernel volumetric convnet modernizing hierarchical transformer for medical image segmentation." arXiv, 2022.
>
> [4] Hatamizadeh, Ali, et al. "Swin unetr: Swin transformers for semantic segmentation of brain tumors in mri images." MICCAI workshop, 2021.
>
> Thank you once again for your valuable insights! (＾ω＾)

---

> > ### Comment · Reviewer_Lrjd · 2024-08-13
> >
> > Thank you for the detailed response and the additional experimental validation. I have raised my score accordingly. Please ensure to add error bars to all tables in the updated paper. Please also conduct tests of statistical significance.

---

### Author Rebuttal · Authors · 2024-08-06

Dear Reviewers,

The authors extend profound gratitude for the meticulous time and expertise invested in reviewing our manuscript. The invaluable insights and critiques provided have been pivotal in guiding comprehensive revisions. In response, substantive modifications have been executed, meticulously addressing all raised concerns in a scholarly manner. Specifically, to Reviewers Lrjd and WWao, we earnestly request a reevaluation of your decisions, predicated upon the thorough efforts undertaken to address your key comments.

In the ensuing responses, each query and concern is meticulously addressed. Numerous extra experiments were conducted, and a PDF comprising the resultant figures has been appended. Table 1 elucidates the performance of D-LKA-former and UNETR++ across four datasets (FLARE2021, FeTA2021, OIMHS, AbdomenCT-1K), providing a comprehensive comparison with attention-based methods. Table 2 assesses the impact of various slicing approaches on model segmentation performance, demonstrating the efficacy of 2D slicing augmentation in enhancing performance across any plane. Figure 1 visualizes the validation curve with Dice Score for 3D U-Net, 3D U-Net+3D uC, and 3D U-Net+2D uC during training, revealing faster convergence and higher performance ceilings with the addition of 2D uC. Tables 3 and Figure 2 delineate the performance across different channel configurations of 3D U-Net, 3D U-Net+3D uC, and 3D U-Net+2D uC on the OIMHS dataset. By substituting the 2D uC module with 3D uC, a performance evaluation of 2D versus 3D convolutions within the same structure is enabled, uncovering better feature extraction capability of 2D convolutions. Image reconstruction experiments were also conducted, and Table 4 evaluates the slice-wise image reconstruction performance of the three models, where the 2D uC model outperforms all 3D convolution-based structures, substantiating the enhancement of planar features by 2D convolutions. The evaluation of the BTCV standard dataset was incorporated to further substantiate the robustness of the proposed method, with results depicted in Table 5. Additional experiments were conducted, as shown in Table 6, to validate the robustness of the DFi conclusions.

These comprehensive experiments collectively underpin the argument that 2D convolutions exhibit good feature extraction capabilities in every plane compared to 3D convolutions. The utilization of uC structured convolutions demonstrably enhances performance, with planar feature augmentation consistently yielding improved outcomes. Given the richness of information on image planes, employing 2D uC for planar enhancement attains optimal performance.

Once again, we express our sincere appreciation for your valuable contributions to the review process. Your expertise and guidance have significantly enhanced the quality of our work and the robustness of our conclusions.

---

### Decision · Program_Chairs · 2024-09-25

**Decision:**

Accept (poster)

**Comment:**

This paper provides a an approach to utilize 2D features via 2D skip connections for 3D CNN U-net segmentation.
The reviewers all appreciate the paper, in particular the simplicity of the idea combined with convincing experiments.
The authors should take care to include their code in the final paper, as well as to utilize the detailed feedback from the reviewers to improve their paper.